# CTF-based soft touch actuator for playing electronic piano

Manmatha Mahato [1,2], Rassoul Tabassian[1,2], Van Hiep Nguyen[1], Saewoong Oh[1], Sanghee Nam[1], Won-Jun Hwang[1] & Il-Kwon Oh [1✉]

In the field of bioinspired soft robotics, to accomplish sophisticated tasks in human fingers, electroactive artificial muscles are under development. However, most existing actuators show a lack of high bending displacement and irregular response characteristics under low input voltages. Here, based on metal free covalent triazine frameworks (CTFs), we report an electro-ionic soft actuator that shows high bending deformation under ultralow input voltages that can be implemented as a soft robotic touch finger on fragile displays. The as-synthesized CTFs, derived from a polymer of intrinsic microporosity (PIM-1), were combined with poly (3,4-ethylenedioxythiophene)-poly(styrenesulfonate) (PEDOT-PSS) to make a flexible elec-trode for a high-performance electro-ionic soft actuator. The proposed soft touch finger showed high peak-to-peak displacement of 17.0 mm under ultralow square voltage of ±0.5 V, with 0.1 Hz frequency and 4 times reduced phase delay in harmonic response compared with that of a pure PEDOT-PSS-based actuator. The significant actuation performance is mainly due to the unique physical and chemical configurations of CTFs electrode with highly porous and electrically conjugated networks. On a fragile display, the developed soft robotic touch finger array was successfully used to perform soft touching, similar to that of a real human finger; device was used to accomplish a precise task, playing electronic piano.

[1] National Creative Research Initiative for Functionally Antagonistic Nano-Engineering, Department of Mechanical Engineering, Korea Advanced Institute of Science and Technology (KAIST), 291 Daehak-ro, Yuseong-gu, Daejeon 34141, Republic of Korea. [2]These authors contributed equally: Manmatha Mahato, Rassoul Tabassian. ✉email: ikoh@kaist.ac.kr

The expeditious development of new technologies demands highly compatible robotic systems capable of doing more advanced and complicated tasks. Although automation and robotics are becoming ubiquitous in all fields of industry, robots for certain delicate applications face serious obstacles. Correspondingly, a new field has emerged in recent years, called soft robotics, to resolve existing and upcoming issues like safety and compatibility. Soft robots have a great potential to be utilized in many current technologies such as wearable devices, drug delivery, advanced surgical tools, soft human-machine interactions, etc. However, to match the target functionalities, the components employed in soft robots must have specific properties such as compliance, flexibility, softness, durability, and material compatibility. The essential component of soft robots is a compliant actuator. Recent advancements in material science have taken advantage of the extraordinary properties of emerging nanomaterials to spur the invention of a new generation of actuators. Until now, a variety of novel soft actuators have been introduced such as piezoelectric actuators, dielectric elastomers, ionic actuators, magnetic actuators, photo-responsive actuators, and others[1]. Among these, ionic soft actuators have shown promising capabilities that can profoundly contribute to solving the existing issues of soft robotics[2–6].

Electro-ionic soft actuators, a type of artificial muscles, are a new class of soft actuators operating based on the movement of unequally sized ions inside an electrolyte membrane. When an electric field is applied to the electrodes, cations and anions move toward opposite electrodes along the thickness direction. Due to the size discrepancy of cations and anions, an asymmetric volume change occurs, leading to bending flexure of the actuator. Therefore, expansion and shrinkage happen mainly by the intercalation and de-intercalation of dissimilar movable ions inside the electrodes due to electrostatic force. The whole actuator works similarly as a capacitor, and numerous studies have been conducted to introduce different electrode materials for ionic actuators capable of storing high levels of charges even under ultra-low input voltages (<1.0 V)[7–14]. Many attempts have been made using carbon nanotubes (CNT), graphene, and graphene-CNT hybrid or composite electrodes to achieve anticipated actuation performance in view of bending deformation and blocking force[15–20]. Wei Chen et al. proposed implementation of graphdiyne as a high capacitive electrode material in ionic actuators. It showed high capacitance of 237 F g$^{-1}$ and generated 16 mm displacement under 2.5 V[21]. Il-Kwon Oh et al. synthesized a carbon-based 3D hetero-nanostructure electrode with exceptionally high specific capacitance of 325 F g$^{-1}$[22]. They claimed that the nitrogen-rich 3D nanostructure provided more accessible ion channels and much larger active surface leading to bending deformation of 6 mm under ultra-low AC voltage of 0.5 V. Recently, Il-Kwon Oh et al. reported an MXene-based ionic soft actuator using Ti$_3$C$_2$T$_x$ as electrode material[23]. Taking advantage of the remarkable properties of the MXene material, they were able to obtain a notably high peak-to-peak bending strain under 1.0 V AC input. For much faster charge transfer, the actuator electrode requires desirable conductive path along with the capacitive material. While many groups have mixed the active material with conductive polymers, Tabassian et al. proposed using a graphene mesh as a conductive network; this mesh was merged with high capacitive nitrogen doped graphene, resulting in an enhancement of actuator performance of more than 600%[24]. Besides the ultra-low driving voltage and high bending deformation, another important factor, which makes ionic soft actuators stand out among their counterparts, is the easy fabrication process. This type of soft actuator can be readily prepared by simple drop casting of electrode material on the electrolyte membrane, and the process is cost-effective considering mass production of real robotic devices. Furthermore, by adding an extra mesh electrode, ionic soft actuators are easily able to self-sense their own movement, which is advantageous with regard to the control systems of future robots[25].

In spite of tremendous potential of ionic soft actuators, they have not been properly employed in soft robots due to certain critical issues hampering their implementation in real applications. Practical applications demand multiple functionalities that are in some cases antagonistic. The key challenge for every component to reach the practical level is to meet all the requirements including durability, low cost, high conductivity, high capacitance, softness, force generation capability, and appropriate mechanical properties. Until now, most studies of ionic actuators have focused on one single specific property, while ignoring most others. For example, it has now been well established that heteroatom-doped porous carbon electrodes, because they can provide high electrolyte-accessible surface area and heteroatom-induced superior surface charge capacitance, can play a significant role in the enhancement of actuation performance of an ionic soft actuator[22,26–29]. However, porous carbon structures derived mainly from metal organic frameworks (MOFs) and covalent organic frameworks (COFs) have lost their inherent chemical structures during carbonization at temperature higher than 400 °C because of thermal instability. Although some organic counter parts are converted into graphite-like short-range structures and slightly increased the electrical conductivity after carbonization, those materials are lacking of long-order π-conjugated frameworks. Therefore theoretically, the stored ionic charges on electrode surfaces do not delocalize as quickly as shown in a fully conjugated electronic long-range system, like CTFs, during charge-discharge cycles, resulting in smaller order of phase delay and relatively lower actuation speeds. In addition, the lack of extended long-order electronic conjugation can put a limit on saturated bending displacement and also can produce certain back-relaxation during long-term exposure to direct current inputs.

CTFs are a new class of nanoporous carbon material having high nitrogen content on their electronically conjugated extended organic frameworks; easily synthesized under solvent-free, cost-effective ionothermal conditions by trimerization of aromatic nitriles (–C≡N)[30]. The nitrogen-containing stable conjugated nanoporous structure, along with very high specific surface area, has made CTFs a favorable material in varieties of advanced applications, including heterogeneous catalysis[31], gas storage[32], electrocatalysts for oxygen reduction reactions (ORR)[33], batteries[34], and high performance supercapacitors[35]. Recently, tetracyanoquinodimethane-derived conductive microporous CTFs containing 8.13% nitrogen were reported for energy storage materials, showing specific capacitance of 383 F g$^{-1}$ in 1 M aqueous KOH solution, high surface energy, and good cyclic stability[36]. The high capacitance value of CTFs is mainly due to the combination of large surface area and heteroatom-induced conductive surface charges. This implies that the choice of aromatic nitrile derivatives as precursors of CTFs is ultimately effective in enhancing the electrochemical performance of the corresponding electrodes. One major advantage of using CFTs in real applications is to retain their inherently stable chemical structures and extended conjugated frameworks in different environments such as open air, common organic/inorganic solvents, and even in harsh conditions such as elevated temperature, acidic or basic solutions, etc., in which they show no compromise of their physical or chemical properties. This robustness of CTFs can be potentially beneficial for improving actuation performance, as well as increasing the durability and lifetime of ionic soft actuators in real applications. However, it may be possible to further improve the electrochemical properties of CTFs by choosing a

proper precursor having additional heteroatom content and permanent microporosity. The polymer of intrinsic microporosity (PIM-1) pioneered by Budd[37] be a perfect precursor for the synthesis of novel CTFs having both of porosity and surface charge capacitance. PIM-1 is a well-known material due to its unique microporous structure originated from inherent rigid spiro-linkage and macromolecular fuse-ring configurations that can resist to pack efficiently in solid state[37], resulting in permanent microporosity on it[37]: it has both nitrogen and oxygen heteroatoms on its backbone and can easily dissolve in common organic solvents like $CHCl_3$, THF, etc. As a result, PIM-1 can be explored for the simple solution casting fabrication of free-standing films, for potential use in selective membrane applications such as purification of gases and liquids[38,39]. Besides its unique microporous structure, PIM-1 has aromatic nitrile functional groups that are highly sensitive to chemical reagents and show great potential for synthetic diversification[40,41]. Varieties of functionalized microporous structures can be chemically synthesized by modifying the nitrile functional groups of PIM-1 and the resulting materials will have unique characteristics in terms of surface porosity and excellent physicochemical stability. Also, such modified PIM-1 materials show excellent electrochemical catalysis properties and can induce separation-adsorption of gas and/or liquid mixtures in ambient condition[42–45].

Here, by utilizing aromatic nitrile groups of PIM-1 as a precursor to achieve hierarchical porous structures with high nitrogen and oxygen content, we report the synthesis of novel conductive CTFs. The synthesized CTFs were utilized as active electrode materials to make an ionic soft actuator for implementation in real field soft robotic applications. The permanent microporosity and extra surface oxygen content of PIM-1, retained in the final CTFs structures, provide extra benefits for soft robotic applications. Permanent microporosity originating from its spirobisindane backbone units gives a large platform for stress-free movement of charged ions; dibenzo-$p$-dioxine moieties present in conjugation with cyclic triazine frameworks serve as charge storage pockets that can boost the specific charge capacitance. These distinctive qualities make it differ from previously reported CTF structures[36]. Finally, hierarchical porous CTFs were used in combination with PEDOT-PSS for flexible electrodes in high-performance ionic soft actuators. The as-fabricated actuator showed exceptional actuation performance in terms of bending displacement under ultralow electric potential, with robust durability of 99% after 15,000 cycles. Furthermore, the generated displacement of the actuator linearly increased with increase in input voltages, which is crucially important considering the design of control systems for real-field robotic applications. The capability of the proposed CTF-based actuator was also successfully validated by demonstrating a soft robotic touch finger that was able to perform a soft touch, allowing it to confidently play music on demand. To the best of our knowledge, CTF-based electrodes have not been employed in the field of soft actuators and the current study can open a new vista to making high-performance and durable ionic soft actuators for soft robotic applications.

## Results

**Synthesis and structural characterization of PIM-1 based CTFs.**
As depicted in Fig. 1a, the ionothermal method was used to synthesize the porous CTFs from PIM-1. To achieve optimum structural properties of the covalent triazine frameworks, ionothermal process was carried out at 400, 500, and 600 °C using 1:5 molar ratio of nitrile linkers of PIM-1 to anhydrous $ZnCl_2$; for simplicity, the obtained CTFs materials are designated as TP4, TP5, and TP6, respectively. All these CTFs were synthesized using

molten dry $ZnCl_2$, which served as both of Lewis acid catalyst and solvent during the reactions. As can be seen in the chemical structure in Fig. 1a, the final triazine framework is a 3D conductive porous-network enriched by nitrogen and oxygen atoms; this structure can boost the charge storage and transfer in 3D space. More detailed descriptions of the synthetic route of the CTF materials are provided in "Methods" section. The prepared CTFs were utilized to make ionic soft actuators for implementation in a soft robotic touch fingers array (Fig. 1b), which requires high-performance, reliable actuators. As can be seen in Fig. 1b, the synthesized CTFs, which were used as an electrode material, can significantly improve the performance of the actuator due to their high electrical conductivity and ultra-high charge storage capacity originating from nitrogen and oxygen-rich porous organic frameworks. In addition, the large electrolyte accessible surface area of the CTFs can accommodate more ions, facilitating swift ion diffusion inside the electrode and resulting in much faster and larger flexural bending deformation under very low input electric fields.

Using various characterization techniques, the structural configurations of synthesized CTFs were carefully investigated. Proton nuclear magnetic resonance ($^1$H NMR) analysis shows that the chemically produced initial PIM-1 materials had sufficient purity before being used for the synthesis of CTFs (Supplementary Fig. 1a). Brunauer, Emmett and Teller (BET) $N_2$ adsorption-desorption isotherm (Type-I) and surface area data (787 $m^2\,g^{-1}$), as shown in Supplementary Fig. 1b, additionally confirm the intrinsic microporous nature of PIM-1. The main goal of synthesizing CTFs based on PIM-1 is to utilize the intrinsic microporosity of PIM-1, while enhancing the electrical conductivity due to the formation of electronically conjugated long-order covalent triazine frameworks and high specific capacitance due to the generation of hierarchical porous heteroatom-containing polar surfaces. The porous structures help to increase the ionic diffusion and electronically π-conjugated long-order triazine frameworks increase the electrical conductivity. Therefore, the reported CTFs having highly porous π-conjugated network structures and heteroatom contents facilitate both of easier ion diffusion and higher electrical conductivity; the enhanced specific capacitance provides higher charge storage functionality, which is crucial for developing high-performance electro-ionic soft actuators. The increased concentration of electron-rich nitrogen in the triazine frameworks, along with oxygen, can potentially enhance the surface charge density and directly contribute to the electrical conductivity and charge storage capacity. The increase in nitrogen content due to the formation of CTFs is confirmed by elemental analyses; these results are in fairly good agreement with the theoretical calculations. As presented in Supplementary Table 1, the nitrogen content increased from 6% for PIM-1[40] to 9.30, 11.40, and, 10.40% for TP4, TP5, and TP6 CTFs, respectively. The other intriguing point about these results is that by conducting the ionothermal process at higher temperature, larger amounts of nitrogen were embedded in to the final CTF products and, consequently, TP5 and TP6 showed higher values of nitrogen content. However, it was found that the ionothermal process was restricted to the temperature of 600 °C, because degradation of the main backbone of triazine frameworks started above this temperature[46]. This fact is supported by thermogravimetric analyses (TG-DTG), which reveal the decomposition temperature of 670 °C for all synthesized CTFs; none of the samples above that temperature remained stable (Supplementary Fig. 2a). Further to ascertain the decomposition of intrinsic porous framework structures above 600 °C temperature, synthesized CTFs at 700 °C (TP7) by following the same experimental procedure was subjected to thermogravimetric and nitrogen-sorption isotherm

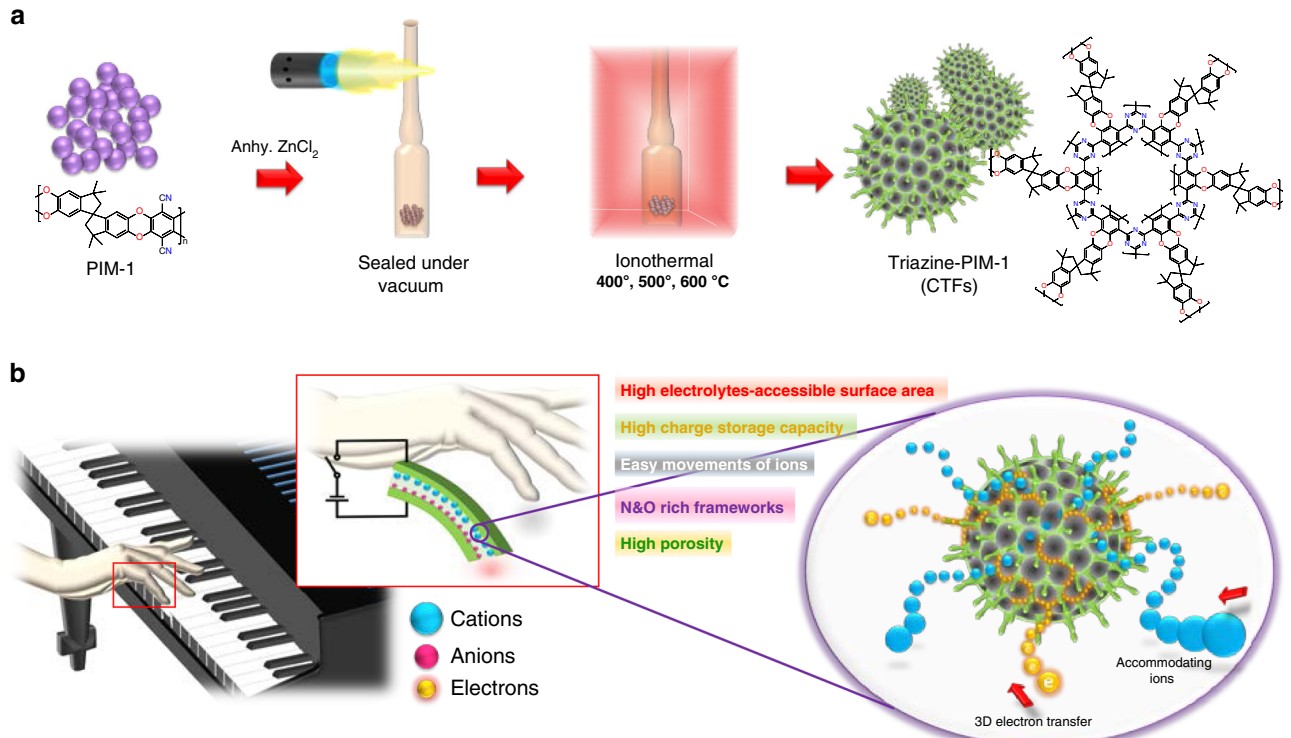

**Fig. 1 Schematic representation of synthesis of Triazine-PIM-1 (CTFs) electrode materials and soft robotic touch finger for playing piano. a** Synthetic routes to polymer of intrinsic microporosity (PIM-1) and porous covalent triazine frameworks (CTFs) at 400, 500, and 600 °C through ionothermal methods. **b** Working principle of soft robotic touch finger and actuation mechanism of ionic soft actuators using CTF electrode materials.

analyses after purification. Absence of major weight loss at elevated temperature further confirms the decomposition of backbone triazine frameworks during ionothermal synthesis of TP7 material (Inset of Supplementary Fig. 2a). It was also observed that the surface area of TP7 is reduced noticeably to 404 $m^2/g$ from 1192 $m^2/g$ (TP6), indicating a major degradation of intrinsic porous back-bone frameworks (Supplementary Fig. 2b).

As expected, scanning electron microscopic (SEM) and high-resolution transmission electron microscopic (HRTEM) images of the samples reveal that the synthesized CTFs perfectly kept the porous nature of PIM-1. The SEM images displayed in Supplementary Fig. 3a–f for TP4 and TP5, and in Fig. 2a–c for TP6 show the porous natures of the CTFs, with well-distributed pores in the whole network-structures. The surface macro-pores would help the access of electrolyte solutions and the movement of ions (cations and anions) upon the application of external electrical potential. The magnified SEM image of TP6 also gives a clear indication of the presence of meso-porosity in the network structures (Supplementary Fig. 4a). The HRTEM images of TP6, shown in Fig. 2d–f, further reaffirm the micro- and mainly meso-porous (>2 nm pore width) nature of the synthesized CTFs, which is beneficial for increasing the electrolyte-accessible surface area[36]. In addition, the HRTEM images shown in Fig. 2f reveal the presence of crystalline and amorphous phases in the TP6 sample. The crystalline regions have been well oriented at the atomic level along both the vertical and horizontal planes (Fig. 2g). For clear visualization, vertically and horizontally oriented crystalline regions were magnified in Fig. 2h and Fig. 2i, respectively.

The dominant porosity and structural integrity of the synthesized CTFs are also well proven by $N_2$ adsorption-desorption isotherm analysis at 77 K. As shown in Fig. 3a, all CTFs showed similar Type-I $N_2$-sorption isotherm behavior that clearly supports their microporous configurations. Specific

surface areas (BET model) of 920, 1071, and 1192 $m^2\,g^{-1}$ were obtained for TP4, TP5, and TP6 CTFs, respectively. These results indicate that not only did CTFs inherit the intrinsic porous nature of PIM-1, but also that their surface area was enlarged compared to PIM-1. The increases of BET surface area due to the formation of CTFs were found to be 133, 284, and 405 $m^2\,g^{-1}$ for TP4, TP5, and TP6, respectively, compared to those of the precursor PIM-1. Pore size distribution of synthesized CTFs was calculated using $N_2$@77-Carb Finite Pores, As = 6, 2D-NLDFT (Non-Local Density Functional Theory) Model, shown in Fig. 3b. Average pore sizes of the TP4, TP5, and TP6 according to $N_2$ isotherms were 2.5, 3.5, and 5.1 nm, respectively, suggesting a mesoporous structure. Micropores (pore size < 2 nm) present in these CTFs (as shown in Fig. 2d–f) were not observed in Fig. 3b, because the pore-size distribution curve is plotted from corresponding nitrogen adsorption-desorption isotherm data of those CTFs. And, it is obvious that adsorption-desorption data were not recorded at very low relative pressure defining micropores during $N_2$-sorption analysis. To check the microporous structures of the synthesized CTFs, argon adsorption-desorption isotherm analysis was performed at 87 K for TP6, as shown in Supplementary Fig. 4b. Rapid uptake of argon at ultralow pressure implies the permanent microporosity of TP6, which exhibits a hysteretic Type-1 sorption isotherm. The NLDFT pore size distribution (Fig. 3b, inset), corresponding to argon physisorption, reveals that TP6 has a dominant micropore size of 1.1 nm, with another two secondary micropores at 1.3 and 1.6 nm. Since PIM-1 derivatives are well-known materials for $CO_2$ adsorption under low pressure[41], the synthesized CTFs under study were also analyzed for the same measurement (Fig. 3c). Isotherms of $CO_2$ adsorption-desorption, shown for TP4, TP5, and TP6, were obtained at 273 K and pressure of up to 1 bar. It is believed that adsorption of $CO_2$ happens due to dipolar interaction with the adsorbent surface and proportional to total surface polarities/

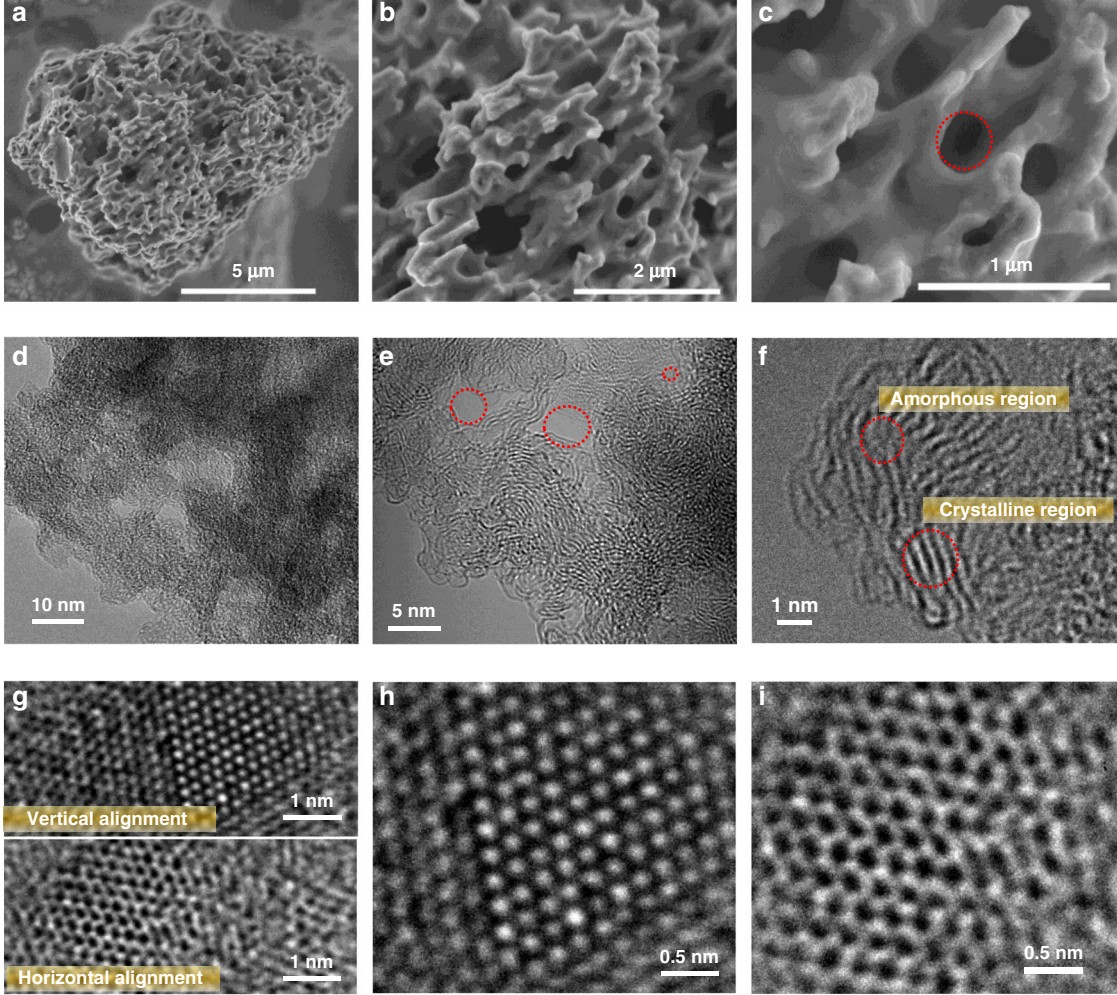

**Fig. 2 Morphological characterization of synthesized hierarchical porous CTFs electrode material at different magnifications. a–c** High-resolution SEM images of TP6 with 5, 2, and 1 μm scale bars showing well-ordered macroporous structures. **d–f** High-resolution TEM images of TP6 with 10, 5, and 1 nm scale bars showing meso and microporous structures. **g** Highest magnified HRTEM images showing both vertically and horizontally aligned crystalline regions of TP6 at atomic level at 1 nm scale. **h** Magnified HRTEM images of horizontally aligned crystalline regions. **i** Magnified HRTEM images of vertically aligned crystalline regions at atomic level at 0.5 nm scale.

charges. Therefore, an increase in $CO_2$ adsorption implies an increase in surface charges. The synthesized CTFs showed much higher $CO_2$ uptake capacity than those of PIM-1 and/or other reported PIM-1 derivatives[40,41]. While TP4 and TP5 showed $CO_2$ uptake capacity of 133.0 mg g$^{-1}$ and 151.0 mg g$^{-1}$ at 273 K and 1 bar, TP6 showed capacity of 152.0 mg g$^{-1}$. Comparing these values with that obtained from PIM-1 under the same conditions (111.4 mg g$^{-1}$)[40] shows improvements of 19%, 35%, and 36% for TP4, TP5, and TP6 CTFs, respectively. The enhancement of $CO_2$ uptake capacity, i.e., surface charge, is due to the combined effect of the increase in nitrogen content and the enlargement of the specific surface area in the synthesized CTFs compared with the initial PIM-1 which has direct impact to the increase of specific capacitance in actuator configuration.

To further investigate the chemical state of PIM-1 based CTFs, spectroscopic characterizations were performed as summarized in Fig. 3d–i and Supplementary Fig. 5 to 7 (see more detail on spectroscopy measurements in "Methods" section). It is clearly observed in Fourier-transform infrared spectroscopic (FT-IR) spectra that the –C≡N groups of PIM-1 underwent CTF formation through ionothermal process at different temperatures, because the characteristic FT-IR peak of PIM-1 at 2241 cm$^{-1}$, corresponding to stretching frequency of –C≡N functional

groups, was not visible in TP4, TP5, or TP6 CTFs (Fig. 3d). This observation clearly indicates the higher order of trimerization reaction during ionothermal process[46]. In addition, a broad FT-IR peak is observed in all CTFs in the range of 1622–1490 cm$^{-1}$, corresponding to –C=C– and –C=N– stretching frequencies, instead of a sharp peak as obtained for PIM-1 at 1620 cm$^{-1}$ due to –C=C– stretching only. The observed peaks at ~1560 cm$^{-1}$ and ~1180 cm$^{-1}$ for CTFs correspond to the vibration of benzene/triazine units[36]. FT-IR transmittance peaks as observed in between 2800 and 3000 cm$^{-1}$ are related to stretching vibrations of –C–H functional moiety whereas the obtained peak at 870 cm$^{-1}$ is corresponding to out-of-plane bending vibrations of it as present in all CTFs materials including the precursor PIM-1. To clearly ascertain those FT-IR peaks corresponding to asymmetric and symmetric stretching vibration of –C–H functional moiety, vibrational frequency region (3100 to 2600 cm$^{-1}$) of it is enlarged (Supplementary Fig. 5). The obtained characteristic FT-IR peaks at 2970 cm$^{-1}$, 2920 cm$^{-1}$, and 2850 cm$^{-1}$ are mainly due to the =C–H ($sp^2$) stretching, –C–H ($sp^3$) asymmetric and symmetric stretching vibrations, respectively. The appearance of those C–H moieties in the FT-IR spectra supports the retained frameworks structure on the reported CTFs. The integrity of framework structures of TP4, TP5, and TP6 CTFs

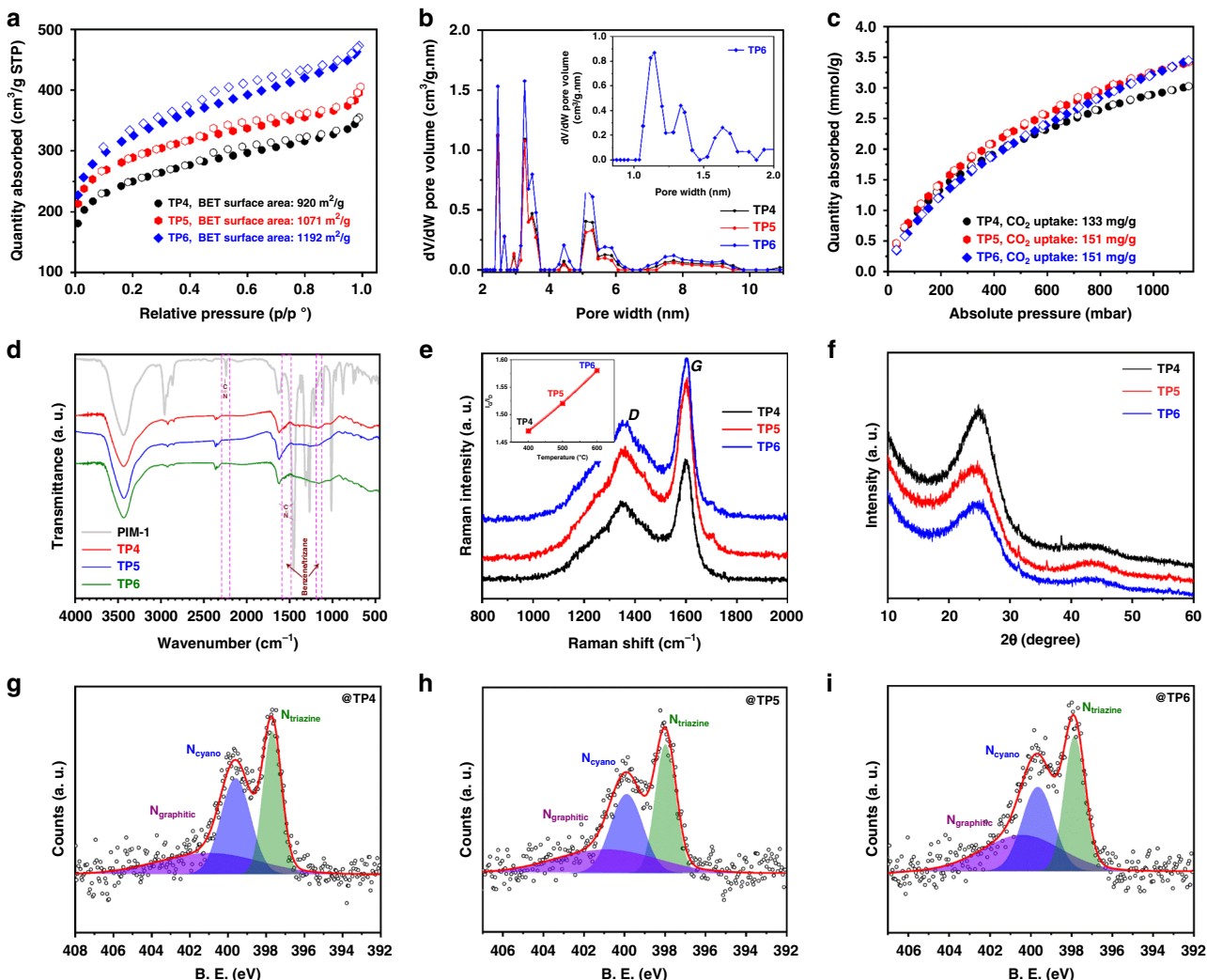

**Fig. 3 Physicochemical and structural characterizations of as-synthesized novel CTFs electrode materials. a** Nitrogen adsorption-desorption isotherms of TP4, TP5, and TP6 at 77 K. Filled and empty symbols denote adsorption and desorption, respectively. Inset: respective specific surface area data. An increase in the surface area from TP4 to TP6 increases the surface energy for physicochemical interactions. All measurements were carried out using 200 mg of each CTF, as mentioned. **b** NLDFT pore size distribution of TP4, TP5, and TP6 electrode materials corresponding to nitrogen isotherm. Micropore (size < 2 nm) is not observed because BET isotherm measurement was taken using nitrogen gas and adsorption-desorption data had not been recorded at very low pressure defining micropores during $N_2$-sorption isotherm analysis. Inset: NLDFT micropores distribution of TP6 corresponding to argon adsorption-desorption isotherm at 87 K (Supplementary Fig. 4b). Pore size < 2 nm confirms the permanent microporosity nature of CTFs. **c** Carbon dioxide ($CO_2$) gas adsorption-desorption isotherms of TP4, TP5, and TP6 materials measured up to 1 bar at 273 K. Inset: respective $CO_2$ uptake data. **d** FT-IR spectra of PIM-1 and corresponding novel PIM-1 based CTFs. Inset: disappearance of nitrile (–C≡N) peak of PIM-1 due to trimerization; appearance of –C=N– peaks due to formation of covalent triazine frameworks (CTFs: TP4, TP5, and TP6) and presence of –C–H moieties in both of PIM-1 and CTFs. **e** Raman spectra of TP4, TP5, and TP6 CTFs. Inset: variation of $I_G/I_D$ with temperature corresponding to TP4 to TP6. Increase of ratio ($I_G/I_D$) confirms the increase size of condensed carbon frameworks/graphitic structures with increase in temperature from TP4 to TP6 electrode materials. **f** High-resolution P-XRD patterns of TP4, TP5, and TP6 show presence of semicrystalline structures. **g–i** XPS N1s spectra of (**g**) TP4, (**h**) TP5, (**i**) TP6 materials show three N configurations after deconvolution as olive: $N_{triazine}$, blue: $N_{cyano}$, and violet: $N_{graphitic}$. Increase in $N_{graphitic}$ peak area from TP4 to TP6 reveals frameworks containing graphitic nitrogen increase with increase in synthetic temperature.

is also confirmed from solid-state $^1H$ and $^{13}C$ cross-polarization magic angle spinning (CP-MAS) NMR spectroscopic analyses as shown in Supplementary Figs. 6 and 7, respectively. The solid-state proton ($^1H$) CP-MAS NMR spectra of PIM-1, TP4, TP5, and TP6 CTFs are shown in Supplementary Fig. 6. PIM-1, which is used as a precursor for the synthesis of CTFs, shows three distinctive proton NMR signals in terms of chemical shift ($\delta$) at 6.40 ppm for deshielded aromatic protons, at 3.80 ppm for –$CH_2$ protons, and at 0.96 ppm for –$CH_3$ protons. In addition, the spectrum consists of broad unwanted signals those are commonly referred as spinning sidebands (defined with asterisks) and

usually originated from the background signals during solid-state NMR analysis. Presence of those assigned protons in the framework structures of TP4, TP5, and TP6 CTFs is also reflected in their solid-state proton ($^1H$) CP-MAS NMR spectra. However, due to change in chemical and physical environments in CTF structures, hydrogen bindings appear at relatively lower $\delta$ values (more shielded) of 5.23, 3.33, and 0.88 ppm corresponding to aromatic protons, –$CH_2$ protons, and –$CH_3$ protons, respectively. Strong intense NMR signal of TP4, TP5, and TP6 at 3.33 ppm for –$CH_2$ protons reveals the structural integrity of the reported CTFs. As expected, the intensity of NMR signal for

aromatic protons at 5.23 ppm decreases from TP4 to TP6 due to increase in the degree of carbonization with the increase of synthetic temperatures from 400 to 600 °C. The –CH₃ bonds show broad NMR signal at 0.88 ppm in all of the reported CTF polymers. In conclusion, the interpreted NMR data proves the presence of hydrogen in the CTF structures in spite of their high synthetic temperatures, elucidating the retention of frameworks although there is a partial carbonization. Solid-state $^{13}$C CP-MAS NMR spectroscopic analysis as shown in Supplementary Fig. 7 also revealed the retention of frameworks by showing the characteristic NMR signals, corresponding to six different types of chemically non-equivalent carbons as present in the reported structures of CTFs (details of $^{13}$C NMR analyses are represented in the Supplementary Information section).

The Raman spectra of TP4, TP5, and TP6 CTFs are presented in Fig. 3e, where it can be seen that the broad $D$ and $G$ bands at 1354 and 1604 cm$^{-1}$ are similar to the fused aromatic cluster in disordered carbon structure[47], confirming the presence of condensed triazine frameworks and benzene rings[36] in synthesized CTFs. The intensity ratios of $G$ and $D$ bands ($I_G/I_D$) for TP4, TP5, and TP6 CTFs also increase with the rise of temperature from 1.47 to 1.58, indicating that the size of the condensed aromatic frameworks increases from TP4 to TP6 (Fig. 3e, inset). The high-resolution powder X-ray diffraction pattern shows two broad peaks at 25° and 44° ($2\theta$) as shown in Fig. 3f. These broad peaks suggest the semi-crystalline nature of the synthesized CTFs[48]. Broadening of these P-XRD peaks corresponding to decreases of crystallinity nature that is also observed from TP4 to TP6 materials as the size of the disordered carbon structure increased with the increment of synthetic temperature. Nitrogen configurations of the synthesized TP4, TP5, and TP6 CTFs were evaluated by X-ray photoelectron spectroscopy analysis, with results shown in Fig. 3g–i, respectively. As expected, three different N1$s$ XPS peaks are observed at binding energies of 398.1 eV for triazine nitrogen (N$_{triazine}$), 399.8 eV for cyano nitrogen (N$_{cyano}$), and 400.7 eV for graphitic nitrogen (N$_{graphitic}$)[36]. The increase in N$_{graphitic}$ peak area from TP4 to TP6 reveals the accretion of graphitic nitrogen-containing frameworks with the increase in synthetic temperature (Fig. 3g–i). This is critically important as the elevation of nitrogen atoms in the frameworks increase the total surface polarities/charges those can profoundly boost the capacitive performance in electrochemical processes when these CTFs are used as electrode materials. In addition, graphitic nitrogen present as a part of CTF structures is beneficial towards the increase of electrical conductivity, which in turn enhances the electrochemical performances and electric double-layer capacitance. Presence of six different types of carbon configurations as stated earlier is also reflected in the deconvoluted XPS spectra of C1$s$ for all of the reported CTFs (Supplementary Fig. 8). The deconvoluted XPS peak at 283.56 eV is related to the carbon attached with hydrogen. The others peaks at 283.86, 284.16, 284.51, 286.09, and 289.58 eV are corresponding to the presence of five different carbon configurations in the CTFs polymer as C–C, C=C, C=N, C=C–C, and C=C–O functional moieties respectively. The existence of those carbons also supports the integrity of CTF structures at higher temperature, up to 600 °C.

The presence of basic elements (carbon, nitrogen, and oxygen) in synthesized CTFs are reflected in the elemental mapping images of TP6 through electron energy-loss spectra (EELS) of carbon, nitrogen, and oxygen (Supplementary Fig. 9a–e). Perfect distribution of carbon, nitrogen, and oxygen was obtained in the overlapped mapping image of TP6 CTFs indicating retention of microporous frameworks in spite of partial carbonization (Supplementary Fig. 9e). The dark regions in the overlapped mapping image correspond to the presence of micro

and meso-pores on the CTFs. The presence of heteroatoms (nitrogen and oxygen) and carbon in the synthesized CTFs structures is also confirmed by X-ray photoelectron spectroscopy (XPS). The XPS survey spectra in Supplementary Fig. 9f show distinct peaks for the carbon, nitrogen, and oxygen elements present in CTFs.

**Electrochemical characterization of synthesized CTFs**. Because the bending deformation of the actuator is directly proportional to the charge storage capacity of the electrodes under certain input electric potential, the electrochemical activity is a key parameter to judge the prospective behavior of an electrode material being used in ionic soft actuators. Here, to evaluate the specific capacitance of the active electrode materials, cyclic voltammetry (CV) testing was performed. To better correlate the CV results with the physical actuation of the actuators, the potential window for CV was fixed in a range from −0.5 V to +0.5 V, as most of the actuation tests were performed within the same input potential window (−0.5−0.5 V). CV measurements were carried out in both aqueous KOH and H₂SO₄ solutions (1.0 M), as well as in non-aqueous EMIM-BF₄/acetonitrile (0.5 M) solution at a constant scan rate of 10 mV s$^{-1}$. The CV responses of TP4, TP5, and TP6 in different electrolyte solutions, as mentioned above, are shown in Fig. 4a–c in terms of current density $vs.$ potential. Interestingly, all CTFs showed almost rectangular-like CV responses, revealing their double-layer capacitance properties. The obtained CV curves are the closest to rectangular shape for the EMIM-BF₄/acetonitrile electrolyte, of which the ionic liquid was employed in the actuator, indicating high stability, together with good electrical and ionic conductivities of the materials in the testing condition. This is vital to prolong the stability of the corresponding actuators and is an important parameter for real applications like soft robotics. In all CV measurements, the areas enclosed by the response curves, corresponding to the charge/discharge capacity, were found to be the highest for TP6 than that of the other two counterparts, TP4 and TP5. The increase in charge/discharge capacity of TP6 was perfectly proportional to the relative surface area of the CTFs. As discussed earlier, TP6 showed the highest surface area, which can provide more space for charge to be stored as well as more accessibility to electrolyte ions, which -lead to higher capacitance. Another intriguing fact about the CV measurements was that the CV curve obtained for the ionic liquid (EMIM-BF₄/acetonitrile) electrolyte was closer to a rectangular shape than were those of the other electrolytes (Fig. 4c). It is particularly important to point out that only EMIM-BF₄ was used as mobile ions in the electrolyte membrane of the ionic soft actuator. Because the ionic liquid was used in the actuator, CV measurements of TP6 in EMIM-BF₄/acetonitrile electrolyte were further conducted at various scan rates from 1 to 10 mV s$^{-1}$ (Fig. 4d). As expected, the areas of the resultant rectangular CV curves increased with increase of the scan rate without deformation of its shape representing the stability of the electrode materials, much needed for long-term ideal performances. From the obtained CV results at 10 mV s$^{-1}$, the average specific capacitance values of all electrodes in different electrolytes for three consecutive cycles of measurements were calculated and are presented in Fig. 4e. The resultant specific capacitances at different scan rates (0.01, 0.05, and 0.1 V/s) corresponding to each cycle of CV measurement in both of aqueous and non-aqueous electrolyte solutions are reported in Supplementary Table 2 and discussed in the Supplementary Information section. Owing to the distinctive porous structural architectures, all materials had very high specific capacitance values, ranging from 200 to 500 F g$^{-1}$ irrespective of the electrolyte solutions. The presence of extra dibenzo-$p$-dioxine moieties in conjugation with

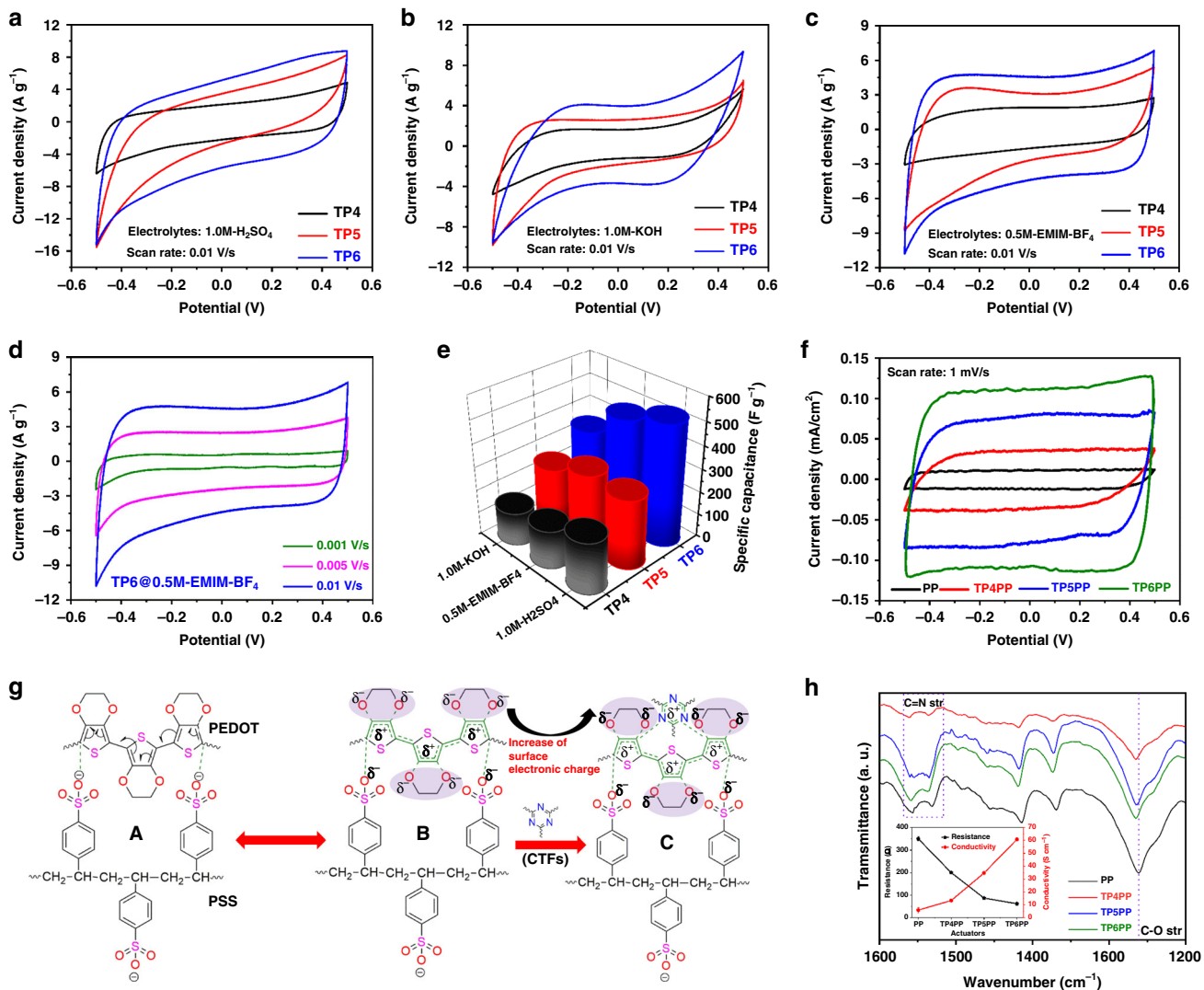

**Fig. 4 Electrochemical performances of synthesized electrode materials and fabricated ionic soft actuators with proposed mechanism.** Cyclic voltammetry (CV) analysis of as synthesized novel CTFs TP4, TP5, and TP6 in **a** aqueous 1.0 M $H_2SO_4$, **b** 1.0 M KOH electrolyte, and **c** non-aqueous 0.5 M EMIM-$BF_4$/acetonitrile ionic-liquid electrolyte solutions at a scan rate of 0.01 V/s. **d** Typical CV response characteristics of TP6 electrode material at various scan rates (0.001-0.01 V/s). **e** Comparative average specific capacitances of TP4, TP5, and TP6, calculated at a scan rate of 0.01 V/s in all electrolytes (Supplementary Table 2). **f** Typical relative CV response patterns for PP, TP4PP, TP5PP, and TP6PP ionic soft actuators fabricated at a constant scan rate of 1.0 mV/s. **g** Probable mechanism for enhancement of actuation performance, explaining changes of structural configuration of PP in presence of synthesized CTFs responsible for high charge storage-discharge capacity. **a**–**c** Electronic interactions involved in increase of surface charge and electric conductivity of PP. **h** FT-IR spectra and changes of electrical conductivity/resistivity (inset), supporting proposed mechanism show clearly the shifting of characteristic FT-IR peaks and increase of electric conductivity of PP upon interaction with CTFs.

cyclic triazine frameworks serving as charge storage pockets boosted the specific capacitance compared to those of other conventional CTFs reported to date[36]. A possible mechanism for the rise of charge storage and accommodation of more ions by dibenzo-*p*-dioxine unit through resonance is presented in Supplementary Fig. 10a and can be explained if we consider condition "A" is the initial configuration of the dibenzo-*p*-dioxine moiety when cations/anions come close to it. The accommodation of cations is shown by Path *a* (A–B) and that of the accommodation of anions is shown by Path *b* (A–C). Configuration "B" shows the ability of the dibenzo-*p*-dioxine moiety to accommodate the cations present in the electrolyte by delocalizing the charges through extended conjugated frameworks toward the cation sites. Configuration "C" does the same for anions.

TP6 showed the highest specific capacitance in all aqueous and non-aqueous electrolytes (Fig. 4e) due to the combination effect

of its higher surface area, richer heteroatom (nitrogen and oxygen) contents and graphitic nitrogen induced high electrical conductivity. To understand the actual charge storage and discharge capability, which are responsible for bending deformation of ionic soft actuators, the CV response for the actuator itself was also measured. For this, four actuators were fabricated using four different electrodes, PP, TP4PP, TP5PP, and TP6PP (PP represents only PEDOT-PSS based actuator). Details of the actuator fabrication process are provided in the "Methods" section. The CV measurements were conducted at a scan rate of 1 mV s$^{-1}$, as shown in Fig. 4f. The obtained rectangular-shape CV response of the TP6PP electrode suggests the perfect double-layer capacitance of this electrode for high-performance actuation at relatively low input voltage. In addition, the TP6PP electrode showed the highest areal capacitance, 114 F cm$^{-2}$, which is more than 11-fold larger than that of pristine PP (10 F cm$^{-2}$). Likewise,

TP4PP and TP5PP actuators showed areal capacitances of 36 and 73 F cm$^{-2}$, respectively, having more than three and seven times larger capacity than that of the pristine PP actuator. The increase of areal capacitance value from TP4PP to TP6PP actuator is also supported by the obtained results of electrochemical impedance analysis as shown in Supplementary Fig. 11 (EIS analysis is represented in details in Supplementary Information section). To check the stability of the charge/discharge cycles, CV responses of the TP6PP-based soft ionic actuator have been measured up to 20 cycles at the same scan rate, as shown in Supplementary Fig. 12a. The rectangular CV responses for all cycles were superimposed, suggesting quite good stability and reversibility of the actuators during charge-discharge cycles. The surprising fact here is that only 0.035 wt% of the CTFs increased the capacitance of PP actuator by many times. It would be interesting to determine whether such massive change in capacitance came from trace amounts of CTFs or from alternation of the PP configuration due to presence of CTFs. It was found that the second is the main reason, as will be explained in the following. During the fabrication process, it was very difficult to disperse CTFs homogeneously in common solvents like water, alcohols, THF, DMF, etc. But, using DMSO as a solvent not only removed the dispersion problem of CTFs, but also, because DMSO and H$_2$O are completely miscible to each other, yielded tremendous compatibility for interaction with aqueous solution of PEDOT-PSS during electrode preparation. In this study, a mechanism is proposed essentially for the alternation of the PEDOT-PSS configuration in the presence of CTFs; this mechanism can easily correlates with the high charge and discharge capacity of CTF-actuators, as shown in Fig. 4g. PSS is flexible and electrical nonconductor, but ionic conductor that not only provides mechanical support to the formation of continuous film with PEDOT, but also acts as dopant by interacting with PEDOT through the sulfonate functional groups, and hence increasing the electronic movement and conductivity in PEDOT (Fig. 4g–a). PEDOT contains ethylenedioxy units which, due to the presence of electronegative ($\delta^-$) O-atoms, play a vital role, together with the sulfonate groups of PSS, in the movement of EMIM$^+$ cations toward both sides of the Nafion membrane. As clarified by Fig. 4g–a, in the presence of PSS, PEDOT acquires electropositive ($\delta^+$) charges by electronic interaction between the O-centre of the sulfonate groups of PSS and the C-centre of the thiophene groups of PEDOT. Therefore, the electronegativity ($\delta^-$) of the oxygen of the ethylenedioxy units decreases, stabilizing the electropositive thiophene units of PEDOT as shown in Fig. 4g–b; correspondingly low areal capacitance is observed for the PP actuator. As it decreases the overall surface charge or polarity and lowers the charge storage/discharge capacity.

CTFs have a very high density of conjugated triazine units, which contain –C=N– moieties where $\delta^-$ charge is stored at the N-centre (acting as donor site) and corresponding $\delta^+$ charges at the C-centre (acting as acceptor site) because nitrogen is more electronegative than carbon. In the presence of CTFs on PEDOT-PSS, the $\delta^-$ charged N-centre, because it has higher nucleophilicity than the $\delta^-$ O-centre of the ethylenedioxy unit, dedicates its electron to minimize the electro-positivity of the poly (thiophene) units (Fig. 4g–c). This electron donor property increases with the rise of the surface charges/polarities, which is directly proportional to the relative surface area and heteroatoms content. Therefore, in the presence of CTFs, not only do the $\delta^-$ charges of the O-centre of the ethylenedioxy unit increase, but also this helps to increase the charge conduction path of PEDOT because the resulting $\delta^+$ charge on the triazine unit is readily delocalized through its extended conjugation networks. This proposed mechanism shows how the areal capacitance of PP increases from TP4PP to TP5PP and TP6PP actuators.

FT-IR (ATR mode) spectroscopic analyses and electrical conductivity measurements were performed for PP, TP4PP, TP5PP, and TP6PP electrodes to support the above-described mechanism. All the electrodes were prepared at identical conditions and constant ratios, with resulting thicknesses of 17–20 μm. As shown in the FT-IR results in Fig. 4h, the stretching frequency of the C–O functional group of the ethylenedioxy unit of PEDOT and the –C=N of the triazine unit of CTFs slightly shifted toward higher wavenumbers than their normal values. For example, C–O bond stretching frequency of PEDOT-PSS shifted from 1260 to 1265 cm$^{-1}$ (Fig. 4h) in presence of CTFs whereas –C=N stretching of CTFs shifted from 1590 to 1495 (Fig. 3d) to 1595–1516 cm$^{-1}$ (Fig. 4h) due to the increase of polarity as a results of electronic interaction in between them. This strongly correlates the increase of $\delta^-$ charges of O-centre of ethylenedioxy unit with the formation of $\delta^+$ charge on the triazine unit. Conductivity data of the as-prepared electrodes showed significant improvement from PP (5.88 S cm$^{-1}$) to the CTFs electrodes, TP4PP (13.32 S cm$^{-1}$), TP5PP (34.71 S cm$^{-1}$), and TP6PP (60.88 S cm$^{-1}$), indicating the electronic contribution of CTFs toward the increase of areal capacitance of the actuators (inset of Fig. 4h).

**Actuation performances of CTF-based ionic soft actuators.** The soft ionic actuators were fabricated by drop casting of electrode solutions on both sides of Nafion/EMIM-BF$_4$ electrolyte membranes. For better comparison, four electrode solutions were casted, as mentioned earlier (PP, TP4PP, TP5PP, and TP6PP). More details on the fabrication steps are provided in the "Methods" section. Using a laser sensor to capture the tip displacement while applying different electric fields, actual bending performances of the fabricated CTF-based ionic soft actuators were investigated in open air. The actuation performance of the synthesized CTF-based actuators perfectly matched the examined physicochemical and electrochemical properties of the samples. As can be observed in Fig. 5a, b, the actuators were tested for square and sine input signals under the same ultralow peak voltage of ±0.5 V and excitation frequency of 0.1 Hz. Although adding CTFs to the PEDOT:PSS increased the stiffness of the electrode (Supplementary Table 3), the positive electrochemical effect of CTFs overwhelmed the obstructive influence of elevated stiffness on bending deformation. For both square and sine wave inputs, the generated bending deformation of the actuators was in the order of PP < TP4PP < TP5PP < TP6PP. Among them, the TP6PP actuator showed predictable high bending deformation. For the square wave input (Fig. 5a), the TP6PP actuator showed a 17.0 mm peak-to-peak displacement that was 3.1 times higher than that of the pristine PP actuator (5.66 mm). In the case of sine wave input (Fig. 5b), the TP6PP actuator showed a 13.5 mm peak-to-peak displacement, 3.4 times higher than that of the pristine PP actuator (4.0 mm) under identical conditions. To the best of our knowledge, this performance is a displacement record, the best value reported to date for electro-ionic soft actuators under ultralow input potential, as shown in Supplementary Table 4. From Supplementary Table 4, it is observed that most of the reported bending displacements of ionic soft actuators is measured by taking the thickness in the range of 80–115 μm. The obtained bending displacement of the TP6PP actuator having the thickness of 115 μm is superior to those of other recently reported actuators based on carbonized heteroatom(s) doped carbon materials derived from COFs[26,28] or doped graphene[22,24,27,49,50]. Likewise, compared with the pure PP actuator, TP5PP and TP4PP actuators respectively showed 220% and 175% enhancements of output displacement for square wave input; and 260% and 190% improvements for sine wave input. The superior

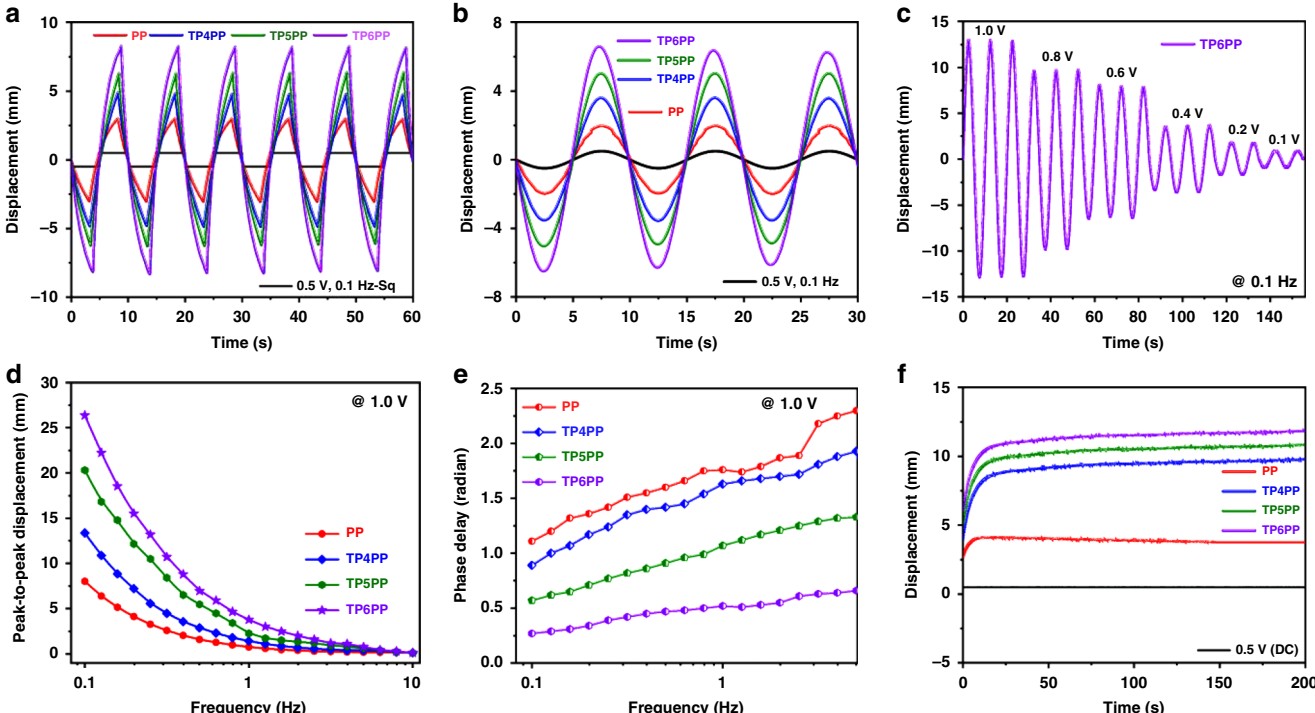

**Fig. 5 Typical actuating performances of fabricated PP, TP4PP, TP5PP, and TP6PP ionic soft actuators. a** Time-dependent bending responses of actuators under ±0.5 V square wave input potential and 0.1 Hz excitation frequency. **b** Time-dependent bending response patterns of actuators under ±0.5 V sine wave input potential and 0.1 Hz frequency. **c** Bending responses of the TP6PP ionic soft actuators with sine wave input potentials varying from 0.1 to 1.0 V, at a constant frequency of 0.1 Hz. **d** Peak-to-peak displacements of PP, TP4PP, TP5PP, and TP6PP actuators with various frequencies from 0.1 to 10.0 Hz at ±1.0 V constant sine wave input potential. **e** Comparative phase delay response patterns of actuators under sinusoidal input potential of 1.0 V with varying of frequency from 0.1 to 5.01 Hz. **f** Time-dependent bending response patterns of ionic soft actuators under ±0.5 V DC input potential.

actuation performances of the TP6PP-based ionic soft actuator are highly correlated to the highest δ⁻ charges of the O-centre of the ethylenedioxy units in the modified PEDOT-PSS structure in the presence of PIM-1 based CTFs, which can act as both electron acceptor and donor. The effective contribution of CTFs to the increase of charge and discharge capacity of PP is further reflected in the linear rise of the bending displacement upon variation of applied input potential from ±0.1 to ±1.0 V, as shown in Fig. 5c. To check the linearity, an error bar plot was made using the displacement data against applied input potentials (Supplementary Fig. 12b). It was observed that the data fitted well linearly with the $R^2$ value of 0.9794. This result strongly supports the idea of the involvement of conjugated structures of CTFs, through which conduction of charge/discharge happens on a linear path without loss of any positive/negative surface charges during the change of applied input potential. It is believed that extended conductive frameworks of CTFs helps significantly for the conduction of applied voltages to obtain surface positive/negative charges that are directly related to the deflection of ionic soft actuators. And, as this conduction happens through extended conductive paths of both CTFs and PEDOT-PSS, the storage charge of active surface increases or decreases linearly with the increase or decrease of applied potential, respectively. This is an exceptional case for the use of only conductive materials as active part, as most of the reported soft actuators consist composite of conductive and semi-conductive materials. Presence of semi-conducting materials restricted the conduction of applied voltages to some extent depending on the magnitude of applied voltages. Therefore, during every charge/discharge cycle at different applied voltages, there is an irregular loss of actual applied potentials giving non-linear deflection. This is the first indication of the durability of the long-term bending performance, as

reflected in Supplementary Fig. 12c. Long-term durability is a key challenge for every state-of-the-art device trying to approach practical application levels in areas such as robotics, because short-lived devices are not sufficiently reliable for robotic implementation. As can be observed in Supplementary Fig. 12c, the TP6PP actuator maintains ~99.0% of its bending performance up to 15,000 cycles of continuous actuation under input voltage of ±1.0 V and oscillation frequency of 1.0 Hz in open air. This excellent stability of the actuator can be explained by the high electrochemical stability of CTFs as described previously. The actuation performance of the CTF-based actuators was also checked at oscillatory potential of 1.0 V and showed similar results, as displayed in Supplementary Fig. 12d–f.

The bending displacement of the TP6PP actuator also took the top position among other actuators under a wide range of sinusoidal frequencies from 0.1 to 10 Hz at a constant amplitude of sinusoidal input (1.0 V), as shown in Fig. 5d. The bending displacements of all prepared soft ionic actuators dropped with the increment of the excitation frequency, mainly due to the shortage of available time for charge-discharge. The superior surface energy and conductivity of TP6PP compared to its counterparts increase its rate of charge-discharge processes and makes it the best actuator in terms of bending displacement over all frequencies. For better understanding of the charge-discharge rate, phase delay between input stimulation and output displacement of actuators was evaluated under a wide range of frequencies (0.1–5.01 Hz) and a constant amplitude of applied input potential (1.0 V). As displayed in Fig. 5e, by accurately calculating the time difference between two equivalent points on the output signals (actuator displacement) and the input signals (applied potential), phase delay evaluation was carried out for PP, TP4PP, TP5PP, and TP6PP actuators[24]. The obtained results gave

clear evidence regarding the charge storage-discharge rate of the proposed actuators. PP shows the longest phase delay over all frequencies, with the slowest charge storage-discharge rate due to its relatively low areal capacitance and slow charge transfer. With the incorporation of PIM-1 based CTFs into the PP, the phase delay considerably drops from TP4PP to TP5PP and TP6PP. It was found that the TP6PP actuator showed phase delays 4.1 and 3.5 times lower than that of the PP actuator at frequencies of 0.1 and 5.01 Hz, respectively. The small phase delay of the TP6PP actuator is mainly due to the change of inherent structural configuration of PP in the presence of CTFs having high areal capacitance and electrical conductivity (Fig. 4g–c). Besides supporting to the high conductivity of electrodes, reducing phase delays is also important for real applications of actuators in a design process of control systems for robotics. Bending responses of PP, TP4PP, TP5PP, and TP6PP actuators under applied DC voltage (0.5 V) also showed good performance (Fig. 5f), following the trend of the AC voltage (PP < TP4PP < TP5PP < TP6PP). However, the maximum displacements of all actuators were notably enhanced in comparison to AC current under the same applied input potential of 0.5 V. For example, compared to the case of AC input, under 0.5 V DC input the TP6PP actuator showed 1.4 times higher maximum displacement. The TP6PP actuator delivered the highest bending displacement of 11.92 mm with the application of very low input DC voltage (0.5 V). A reason for these actuation enhancements is that DC stimuli gave sufficient time for the actuators to reach their full bending, but AC stimuli did not. For all actuators in DC voltages, the minimum time to attain maximum bending displacement, or the rise time, was less than 10 s. More importantly, none of the synthesized CTF-based actuators showed back-relaxation during the DC input voltage, which makes them stand out among their counterparts for utilization in robotic applications those require easily designed control systems. It was interesting to observe that in this case of applied DC stimulation (0.5 V), the TP6PP actuator showed an actuation performance 3.2 times higher than that of PP at all instants of time. The maximum blocking force of the TP6PP actuator under DC input was also measured with various amplitudes of the stimulation. As could be seen in Supplementary Table 5, the developed actuator is able to generate relatively high blocking force, 24 times of its weight, under DC input of 2.0 V. Comparing this data with that of PP actuator reveals that the force capability of the CTF-based actuator is improved up to almost 100% by simply adding CTFs to the PP electrode. It means that a light-weight and organic cyclic triazine frameworks are suitable for active electrode materials in a metal-free electro-ionic soft actuator. In addition, there is tremendous opportunities to increase the blocking forces further in near future by increasing the thickness of the actuators or utilizing much stiffer ionic polymer membranes rather than Nafion. Besides, there is a possibility to incorporate some heavy conducting electrode materials or metal nanowires on CTFs, which not only prevent the air-oxidation of metal, but also increase the conductivity and specific capacitance of the electrodes in many folds.

The measured characteristics of the novel CTF-based actuator suggest the wide potential of this actuator for implementation in various soft robotic applications. The important advantages of CTF-based actuators, such as high bending deformation under ultra-low input voltage, along with long durability, short phase delay, linear increment of output displacement, and lack of back relaxation, are all crucial in control systems for robotic applications.

**Demonstration and operation of soft robotic touch finger**. The intriguing results of actuation performance show the promising capability of this newly developed ionic actuator for soft robotics applications. Future robots will need to accomplish many complicated tasks, which will require highly controlled soft touch to deal with sensitive objects; such robots will include surgical robots, agricultural robots, etc. One of the best devices by which the capability of the soft actuators can be evaluated for ability to perform soft touch are touch screens. Fragile touch screens are very sensitive to mechanical force and should be operated by a very gentle and controlled level of loads because a harsh touch by a rigid object can potentially destroy the whole device. The soft touch capability of the developed actuator motivated the design and fabrication of a soft robotic touch fingers array that can be used to play piano in a smart phone application program. An artificial soft robotic fingers array was fabricated using ten soft ionic actuators (TP6PP) to play piano on a touch screen. Schematic for the working principle of playing piano application on a smart phone by a soft touch robotic fingers array was illustrated in Fig. 6a. The actuators were gripped between two sets of electrodes, which were aligned in a row. Figure 6b shows the arrangement of the actuators for the fabricated soft artificial robotic fingers array. A hardware keyboard was also fabricated to activate each of the actuators separately (Fig. 6c). Pressing one of the keys on keyboard closed the circuit for the corresponding actuator and led to the stimulation, as illustrated in Supplementary Fig. 13. The set of actuators was then mounted on a touch screen running a piano App (Fig. 6d). Playing the hardware keyboard eventually stimulated the soft actuators and forced them to touch the corresponding piano keys on the screen. As illustrated in Fig. 6e, all ten artificial fingers worked properly and pressed the piano keys successfully. To better observe the operation of the artificial robotic finger array, see Supplementary Video I. The as-prepared artificial touch finger array was then used to play 'Happy Birthday', as displayed in Supplementary Video II. As can be seen in this video, the ionic soft actuators were sequentially activated to touch the screen, such that the music was nicely played by the soft artificial touch fingers. These promising results demonstrate the capability of ionic soft actuators for practical implementation in a variety of soft robotic applications in the future.

## Discussion

Highly porous structures of semi-crystalline PIM-1 based CTFs play a significant role for the enhancement of actuation performance of electrode materials in ionic soft actuators. The presence of dibenzo-*p*-dioxine moieties in conjugation with conductive CTFs provided a huge platform to increase the rate of charge storage and discharge capability in aqueous acidic and alkaline systems, as well as in non-aqueous ionic-liquid systems. This led TP6 to achieve phenomenal average specific capacitances of 522, 337, and 467 $F\,g^{-1}$ at a scan rate of 0.01 V/s in aqueous acidic, alkaline, and non-aqueous ionic liquid systems, respectively, along with high cyclic stability. Owing to its high surface area and heteroatom (nitrogen and oxygen) content, the proposed TP6 acquired highest surface polarities/charges showing 36% higher $CO_2$ uptake capacity, which is the highest capacity among all reported PIM-1 based materials. Use of a small volume of DMSO provided perfect compatibility between synthesized CTFs and PEDOT-PSS and demonstrated the good capability of CTFs, which showed the best-ever bending displacement of ionic soft actuators under ultralow input voltages. Under input potential of ±0.5 V and excitation frequency of 0.1 Hz, the TP6PP actuator delivered highest peak-to-peak displacements of 13.5 mm and 17.0 mm for sine and square wave inputs, respectively.

The mechanism of the high bending displacement and long-term actuation cyclic stability has been clearly explained as well as

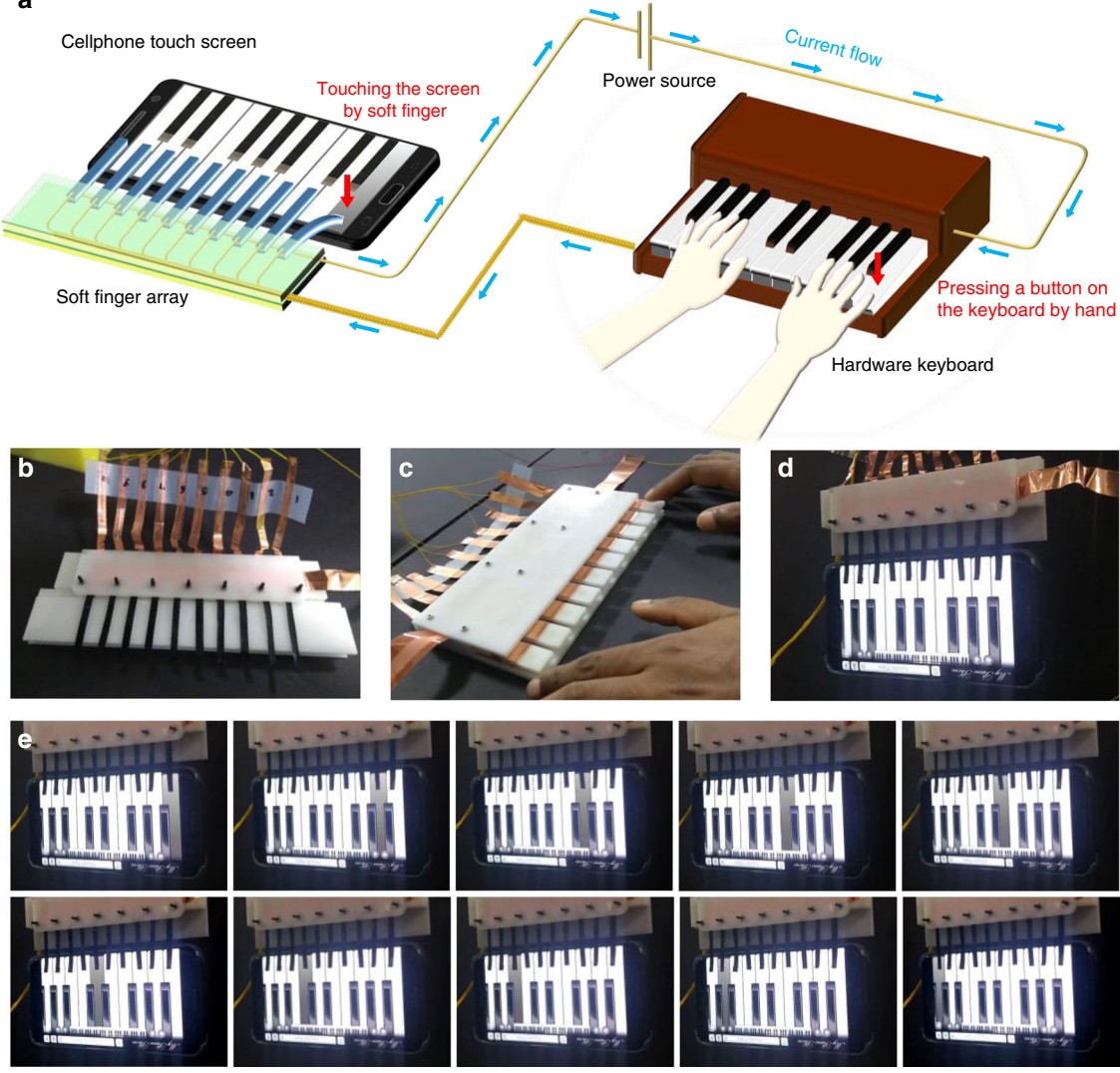

**Fig. 6 Fabrication and operation of soft robotic fingers array to play piano. a** Schematic illustration for the working principle of playing piano application on a smart device by soft touch robotic fingers array based on a lab designed hardware, controlled by human fingers. **b** Chronological arrangement of the air-working ionic soft actuators with respective electrical connections of the fabricated artificial robotic fingers array. **c** The hardware keyboard designed to switch the input stimulation between the actuators. **d** The soft touch fingers array mounted on the touch screen, which is running a piano App. **e** Operation of all soft touch fingers by applying input stimulation to play piano for "happy birthday" song.

validated in the previous section. The presence of CTFs increased the surface charges and charge conduction paths of PP, as supported by the FT-IR and conductivity data. The increase of surface charges was directly proportional to the higher charge-discharge capacity, corresponding to the high bending displacement. Increase of charge conduction paths provided easy delocalization of surface charges without static charge loss, resulting in high cyclic stability without any back relaxation, as can be seen in Fig. 5f. The linear relationship between bending displacement and input potential, in the range of 0.1–1.0 V, matched well with the structural configuration of TP6, which had high surface charge and extended delocalization paths. The delocalization path was able to help to adjust the surface charges by changing the applied input voltages at a constant rate up to the saturated potential. The distinct bending response patterns for PP, TP4PP, TP5PP, and TP6PP actuators under variable input potentials and frequencies matched their conductivity and areal capacitance data well. In addition, the proposed CTFs strongly contributed to reduction of phase delay; the TP6PP actuator had a phase delay up to 4.1 times lower that of the pure PP actuator. The CTF-based

actuator showed no back relaxation under prolonged exposure to DC voltage, and exhibited enhanced bending displacement of up to 11.92 mm under application of ultralow input potential (0.5 V).

Finally, as reported in this study, the controlled and superior bending responses of the TP6PP actuator under ultralow input voltages was able to be utilized for fabrication of soft robotic touch fingers array and other real applications. The fabricated soft touch fingers confidently accomplished a soft touch task, running musical apps in a controlled and methodical way. In addition, this well-design CTF-based soft robotic finger can potentially be used for other complicated applications that demand precise operation, such as activation of sophisticated biomedical devices under areas of restricted human hand use, or operation of certain apps to control experiments in hazardous environments.

## Methods
**Materials**. 5,5′,6,6′-tetrahydroxy-3,3,3′,3′-tetramethyl-1,1′- spirobisindane (>96%), and tetrafluoroterephthalonitrile (>98%) were purchased from TCI, Japan. Anhydrous potassium carbonate ($K_2CO_3$, 99.5%), $N,N$-dimethyl formamide (DMF, 99%), chloroform ($CHCl_3$, 99.5%), methanol (99.5%), acetone, 1,4-Dioxane, and tetrahydrofuran (THF, 99%) were purchased from SAMCHUN, South Korea.

Anhydrous Zinc Chloride (ZnCl₂), Hydrochloric acid (HCl), Dimethyl sulfoxide (DMSO), CDCl₃ (99.8%), and DMSO-d⁶ (99.96%) were purchased from Sigma-Aldrich, USA.

## Experimental

*Synthesis of PIM-1.* Synthesis of PIM-1 was carried out according to the procedure reported by Patel et al.[40] as shown in Supplementary Fig. 10b. In this typical synthetic procedure, a three-neck round bottom flask was initially charged with tetrafluoroterephthalonitrile (2.0 g, 10 mmol), 5,5′,6,6′-tetrahydroxy-3,3,3′,3′-tetramethyl-1,1′- spirobisindane (3.4 g, 10 mmol), and anhydrous N,N-dimethyl formamide (70 mL) solution. The whole reaction mixture was stirred under inert atmosphere (N₂) at 65 °C for 30 min to obtain a clear solution. Then, K₂CO₃ (4.15 g) was added slowly to the reaction mixture and stirring was continued for 72 h under the aforementioned reaction conditions. After cooling to room temperature, the obtained reaction mixture was added to distill H₂O (700 mL) under stirring and the obtained PIM-1 precipitate was collected through filtration. PIM-1 was dissolved in chloroform (200 mL) and precipitated from methanol (500 mL) solution. It was further precipitated by initially dissolving in THF (250 mL) and then adding it to the acetone-THF mixture (600 mL, 2:1 v/v). The precipitated PIM-1 was then vacuum filtered and washed with 1, 4-dioxane (100 mL), acetone (100 mL), water (100 mL), and a further portion of acetone (60 mL). The final product was dried in a vacuum oven at 110 °C overnight to obtain PIM-1 as a luminous yellow powder (Yield: 66.5%, 3.6 g).

*Synthesis of Triazine-PIM-1s (CTFs).* Ionothermal process, which melts anhydrous zinc chloride at high temperature, has been utilized for obtaining the active electrode materials. Nitrile (–C≡N) groups show very good solubility in this ionic melt due to strong Lewis acid-base interactions at the same time ZnCl₂ act as a good catalyst for trimerization reaction to form CTFs[30]. For the synthesis of Triazine-PIM-1s (CTFs), first, an ampule was charged with PIM-1 monomer (100 mg, 0.21 mmol) and 5.0 equiv. of anhydrous ZnCl₂ (147 mg, 1.076 mmol). This was then dried under vacuum at 100 °C for 8 h. Then, respective ampules were flame-sealed under vacuum and slowly heated to the desired temperatures (400/500/600 °C) at a heating rate of 1 °C/min for 48 h in box furnaces. After the furnace oven had cooled to room temperature, the ampule was opened, and the crude material was collected. Black materials obtained were then ground and refluxed in water for 8 h to remove unreacted ZnCl₂, which was filtered and thoroughly washed using hot water. Subsequently, for the removal of unreacted organic impurities from the obtained materials, the black powder was stirred with 1 M HCl solution overnight, followed by successive washings with water until it reached a neutral value (pH = 7). A final wash with acetone was used to remove the water. The materials were then dried under vacuum at 110 °C overnight prior to further analysis. The proposed chemical structure of the as-synthesized CTFs, along with the synthetic procedure, is shown in Supplementary Fig. 10c and corresponding soild-state ¹H and ¹³C CP-MAS NMR spectroscopic analysis for CTFs are shown in Supplementary Figs. 6 and 7, respectively.

*Preparation of soft ionic actuators.* For the study of the actuating performance, the following steps were taken for fabrication of CTF-based soft ionic actuators.

*Preparation of ionic liquid loaded Nafion membrane.* Nafion (0.05 g/mL) in DMAc was dissolved completely by stirring in an oil bath at 60 °C for 10 h. EMIM-BF₄, an ionic liquid (0.03 g/mL), was then added and stirring was maintained for 24 h at 45 °C to form a homogeneous solution. Ionic liquid loaded Nafion membrane with thickness of 80 μm was obtained by casting of the as-prepared homogeneous solution (5 mL) onto a clean and flat glass petri-dish (diameter 5 cm); this was followed by removal of solvent in a vacuum oven at 85 °C for 7 h.

*Preparation of Soft Ionic Actuators.* For the fabrication of the sandwiched structure of ionic soft actuators, an initially homogeneous dispersion of CTFs in DMSO (1 mL, sonication) was made and then added to the solution of PEDOT-PSS (CTFs, 0.35 mg/mL) under stirring. This whole electrode mixture was stirred overnight at 25 °C to ensure that the CTFs were uniformly distributed throughout the PEDOT-PSS and could reach the most favorable level of interaction among themselves. Finally, equal volumes of as prepared electrode mixture were coated on both surfaces of ionic liquid loaded Nafion membranes, one-by-one, through solvent evaporation of 1 h duration at 70 °C. Through this process, three soft ionic actuators of thickness 115 ± 5 μm, namely, TP4PP, TP5PP, and TP6PP, were obtained, corresponding to the CTFs TP4, TP5, and TP6, respectively. Here, the optimized dose of CTFs (0.35 mg/mL) with respect to PEDOT-PSS is determined by checking actuation performance data and homogeneous dispersion of CTFs on PEDOT-PSS solution before coating on to ionic liquid-loaded Nafion surfaces. For comparative measurement of the actuating performances, all ionic actuators were kept at identical size (length and width) and stored in a dry desiccator. For comparative study, a PEDOT-PSS-based ionic soft actuator (PP) was prepared following the same method described above, but in the absence of CTFs.

## Materials characterization

*Structure characterization.* A series of advanced scientific instruments were utilized to characterize the synthetic materials and to explain the contributions of their unique chemical structures towards the enhancement of actuation performance.

Liquid-state ¹H NMR spectra were recorded using CDCl₃ as solvent on a Bruker DMX400 NMR spectrometer. Solid-state ¹H and ¹³C cross-polarization magic angle spinning (CP-MAS) NMR spectroscopic analyses were performed using Solid 400 MHz NMR Spectrometer (Agilent 400 MHz 54 mm NMR DD2). FT-IR spectra were recorded with KBr pellets using a Thermo Fisher Scientific FT-IR spectrometer (Nicolet iS50). Raman spectra were collected using a High-Resolution Raman spectrometer (LabRAM HR Evolution Visible_NIR, HORIBA) using a laser with an excitation wavelength of 532 nm and a laser power of 0.5 mW. XRD patterns were recorded using a High-Resolution Powder X-ray diffractometer (SmartLab, RIGAKU) with Cu-Kα as radiation source (λ = 1.54 Å). X-ray photoelectron spectroscopy (XPS) measurements were carried out using a Thermo VG Scientific Instrument (K-alpha) at base pressure of 3 × 10⁻⁸ Pa. Field Emission Scanning Electron Microscopy (SEM) images were taken from a Hitachi-SU8230 SEM. Field-emission transmission electron microscopy (FE-TEM) images and elemental mapping images were obtained from a Titan Double Cs corrected TEM (Titan cubed G2 60-300, FEI) with STEM-HAADF attachment. Thermogravimetric analyses (TGA) were performed on a NETZSCH-TG 209 F1 Libra High Resolution TGA instrument by heating the samples up to 800 °C at 10 °C min⁻¹ under N₂ atmosphere. Elemental (CHNO) analyses were performed on a Thermo Fisher Scientific FlashSmart) elemental analyzer (CHNS-O). In order to determine the porosity of the powder samples, after the samples were degassed at 150 °C for 12 h under vacuum, nitrogen adsorption-desorption isotherms were obtained utilizing a Micromeritics ASAP 2020 accelerated surface area and porosimetry analyzer at 77 K. To measure the microporous nature of the samples, argon adsorption-desorption isotherms were obtained at 87 K. The adsorption-desorption isotherms were evaluated to obtain the pore parameters, including Brunauer-Emmett-Teller (BET), Langmuir, pore size, and pore volume. Low pressure CO₂ adsorption-desorption isotherms for powder samples were also obtained at 273 K using a static volumetric system (ASAP 2020, Micromeritics Inc., USA). The temperature during adsorption and desorption were kept constant using a circulator. All adsorption-desorption experiments were carried out twice to ensure reproducibility. There were no noticeable differences in the isotherm points of both experiments. The electrical conductivity of the PEDOT-PSS-CTFs composite membrane (thickness 18 ± 2 μm) was measured using a standard four-probe measurement system (MSTECH, Model 4000) in connection with a Keithley SourceMeter 2400. Mechanical properties of Nafion, PEDOT-PSS, PEDOT-PSS-CTF electrodes were measured by using a table-top universal testing instrument (AGS X; Shimadzu Corp., Japan). Blocking forces of the actuator were measured by using LVS-5GA load cell (Kyowa) as a force sensor with capacity of 50 mN according to our previously reported experimental set-up[49].

*Electrochemical characterization.* Cyclic voltammetry (CV) was measured by a conventional three-electrode cell (Ag/AgCl electrode as reference and platinum foil as counter electrode) using a multichannel potentiostat/galvanostat (VersaStat, Princeton Applied Research). The CTF-only electrode materials were prepared by conventional process as reported by Li et al. groups[36]. CTF (80 wt%, 2 mg), acetylene black (10 wt%, 0.25 mg) and polytetrafluoroethylene (10 wt%, 5 μL 5 wt% PTFE) were mixed well through sonication with known volume of solvent (DMSO) and calculated amount of CTFs were loaded on a working electrode (freshly polished glassy carbon electrode). After complete evaporation of solvent, electrochemical measurements were performed. The tests were conducted in aqueous solutions (1 M) of H₂SO₄, KOH, and EMIM-BF₄/CH₃CN (0.5 M) by loading known amounts of TP4, TP5, and TP6 on the working electrode. The CV spectra of all the actuators (PP, TP4PP, TP5PP, and TP6PP) were measured using a conventional two electrode system in which an ionic liquid (EMIM-BF₄) embedded Nafion membrane acted as electrolyte[24]. The specific capacitance (Csp) was obtained in F g⁻¹ from the CV analysis data, using Eq. (1):

$$C_{sp} = \frac{A}{2 \times \Delta V \times v \times S} \tag{1}$$

where, $A$ is the area inside CV curve, $\Delta V$ is the potential window, $v$ is the scan rate, and $S$ is the weight or area taken of sample.

## Measurement of actuation performance.

The actuation performance of the prepared actuators was examined by applying input voltages with various amplitudes and frequencies while recording the tip displacement of the actuator using a laser displacement sensor (Keyence, LK031).

The phase delay (Δφ) corresponding to each ionic actuator under study was calculated by Eq. (2):

$$\Delta \varphi = 2\pi f(\Delta t) \tag{2}$$

Where $\Delta t$ refers to the time delay, which can be obtained by calculating the time differences between two equivalent points on the input potential signal and output displacement signal. The value $f$ represents the frequency of the signal.

## Data availability

All data needed to evaluate the conclusion in the paper are present in the paper and/or the Supplementary Information. Additional data related to this paper are available on request from the corresponding author.

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

## Acknowledgements

This work was supported by Creative Research Initiative Program (2015R1A3A2028975) funded by National Research Foundation of Korea (NRF).

## Author contributions

M.M. and I-K.O. conceived the idea of Novel CTF-based ionic soft actuators. M.M. and R.T. contributed equally to the design of the entire set of experiments, analyzed the data, and wrote the draft of the paper. S.O. and S.N. did experiments on capacitance of the materials for reproducibility and repeatability. V.H.N. and W-J.H. contributed data analyses and revised the paper. I-K.O. supervised the research at all stages, led all groups, and wrote the paper. All authors discussed the results and commented on the paper.

## Competing interests

The authors declare no competing interests.
