## [Peer Review File · Nature Communications]

Reviewers' comments:

Reviewer #1 (Remarks to the Author):

Mahato et al. have used cyclic triazine frameworks to prepare capacitor based soft touch actuators for playing an electronic piano. The work utilizes the high surface area, heteroatom dopants of polymer of intrinsic microporosity (PIM), and thus relatively high capacitance to obtain a displacement of 17 mm under 0.5 V of square wave voltage. The results demonstrate good use of porous materials for actuation. The work, however, has a few fundamental problems that must be addressed.

1. First of all, the authors have completely miscalculated the capacitance. The authors have reported capacitance values in the range of 300-525 F/g for the devices. If the authors have double-checked the literature, they should have realized that the value of 525 F/g is an exceptionally high gravimetric capacitance, substantially higher than many other materials. After carefully checking the section of "Materials and Methods", the reviewer has found that the authors have mis-calculated the capacitance by a factor of two. In other words, equation (1) is completely wrong and the authors have missed a pre-factor of $\frac{1}{2}$. The reviewer is afraid that the authors should not have made a mistake like this for a manuscript submitted to this journal.

2. In the introduction, the authors commented that "porous carbon structures have some limitations in terms of charge accumulation and transfer because they lose extended electronic conjugations during carbonization. Therefore, the stored charges on electrode surfaces do not delocalize quickly during charge-discharge cycles, resulting in [a] high order of phase delay and low actuation speed." This statement is wrong or at least highly questionable. Upon carbonization, carbon atoms are becoming more and more conjugated and eventually graphene-like or graphite-like, resulting in a fully conjugated system with high charge mobility. In fact, for many carbon materials, the capacitance becomes fully electrical double-layer and the phase delay will start to diminish. Thus, the argument that they lose conjugation and result in slow charge/discharge cannot stand (unless the authors have data to prove it or use references to back it up otherwise). The authors may have neglected that, for some porous carbon materials, the phase delay can be caused by reasons such as material mechanical properties.

3. On page 8 and 9, the authors claim that "the highly porous networked structure facilitates easy ion diffusion and high electrical conductivity.....The increased concentration of electron-rich nitrogen in the triazine frame works, along with oxygen, can potentially enhance the surface charge density and directly contribute to the higher electrical conductivity and charge storage capacity". This statement is questionable, and it needs supporting evidence. Can the authors explain why porous networked structure will facilitate electrical conductivity? It is understandable the porous network facilitates ion diffusion, but it is counterintuitive to enhance electron mobility.

4. In the TGA analyses, the authors made a conclusion that the series of TP4, TP5, and TP6 have the same decomposition temperature of 670 C. However, Figure S2 shows contradictory data with completely different conclusions. The decomposition temperatures of these materials are completely different, and they increase in the sequence of TP4, TP5, and TP6. The authors should

have used a scientific protocol to analyze the data properly, for example, the standard 5% weight loss.

5. The N₂ isotherms of TP4, TP5, and TP6 show similar pore sizes of 2.5, 3.5, and 5.1 nm. Can the authors explain or comment on why they show these particular pore sizes? In addition, rapid uptake of argon implies permanent microporosity of TP6. What do the authors mean by “permanent”?

6. The CO₂ isotherms of TP5 and TP6 show similar uptake. Can the authors elaborate the cause of this phenomenon, especially considering their completely different surface areas?

7. In the Raman analysis, the authors have calculated the G and D bands ratios to be in the range of 1.47 to 1.58. Does this mean the materials have become carbon instead of remaining as polymers? In what range of G/D these materials can be still called polymers of intrinsic microporosity?

8. It is highly recommended that authors perform mechanical property analysis of the materials used here. The mechanical properties can play an important role in the actuation and recovery of the ‘fingers’. It is possible, and to some extent, it is impossible to rule out that the mechanical properties can play a more critical role than the surface area and heteroatoms that have induced the increase in capacitance.

9. Lastly, the actuation is mostly investigated at 0.1 Hz. At this frequency, it takes about 10 seconds for each actuation. Will this be relevant in practical use? Will someone wait for 10 seconds for each actuation? As shown in Figure 4d, at 1 Hz the peak to peak displacement becomes drastically smaller. Will this be sufficient for practical use?

Reviewer #2 (Remarks to the Author):

The authors present a variant of ionic EAP actuators like IPMC that are based on double layer charging. In this case it has two PEDOT-PSS electrodes doped with triazine molecules (CTF). Ions migrate between the two electrodes inducing a bending. This kind of actuators is well known and characterised in the field: classical IPMC (see works of Kim), carbon-based materials (works of Aabloo) or conducting polymers (works of Vidal).

The really interesting new novelty of this work is that their actuators do not show the so-called back relaxation, which is a common issue with IPMC actuators.

The title (and focus of the paper) is misleading and should focus on the science, not an obscure demonstration. “Playing the piano” sounds much more than it actually is. The “playing the piano” part is not very impressive nor is it true. The bending actuator array only touches/taps a smart phone, which is chosen to run a piano app. This action does not take any force. All EAP actuators can do this. It is a fun demonstration from the authors, but this fits a demo-session at a conference or a public outreach event to generate interest for high school students.

Also, the title indicates a new type of actuator having a unique soft touch, but most (if not all) ionic actuators have soft touch. This is actually their biggest drawback over electronic DEAs. The lack of useful (blocking) force. If the authors had developed a soft actuator that could play a real

(mechanical) piano, i.e. an EAP actuator that has the stroke and force to press a real piano key, that would have a nice breakthrough worth this journal.

The authors should revise their manuscript considerably. Focusing on the science and downplaying or removing the piano part. Also, the text should be written in a more scientific way and not as a sales brochure. The text is full of sales pitch words e.g. “exceptionally”, “excellent”, “superlative”, etc. Interesting results but they fit a more targeted, narrower audience. A specialised journal on transducers, actuators is recommended for resubmission of the revised text.

Reviewer #3 (Remarks to the Author):

This paper is concerned about a new electrode material of electroactive polymer (EAP) actuator. In this manuscript, the authors successfully fabricated flexible electrodes for ionic EAP actuators by preparing the composites of PEDOT/PSS with synthesized PIM-1 based CTFs which have a porous structure and chemical affinity with ions. Furthermore, they showed that the fabricated actuators had an excellent performance of large displacement and explored a basic mechanism of its performance by analyzing the structure of electrodes perfectly. Hence, I think that this paper will be publishable in Nature Comm. Before publishing, however, if the following points would be revised, I think that this paper must be more useful for the readers in related research fields. So, the authors should be considered and revised the following points:

1. Page 6, line 10: A word, “a polymer of intrinsic microporosity (PIM-1)” is an important key word that shows polymer material. Here, the authors should define what it is by using Figure S6(b) exactly.
2. In the manuscript, the performances of the actuator are evaluated by the displacements. But the displacement is known to depend on the dimension of the actuator film. According to Table S3 that summarizing “Comparison of bending performance of ionic soft actuators”, there are no data of the thickness of each actuator film. It is well-known that the bending displacement increases with decreasing the thickness of the actuator. So, the authors should calculate the dimensionless parameters from the displacements, and discuss the performances of the fabricated actuators.
3. In the manuscript, the authors discuss fast transfer of ions in the electrode in relation to its porous structure. In order to discuss the ionic and electronic transfer of electrodes, the authors should add the electrochemical impedance measurements of fabricated actuators, and discuss the impedance data in the manuscript or the supplementary information.
4. The data of the generated force summarized in Table S4 seems to be not good or usual as compared to that of other ionic polymer actuators previously reported. The authors should discuss the reason for this and how to improve this based on the proposed electrode material in this manuscript.

Reviewer #4 (Remarks to the Author):

In this paper, Prof. Oh reported a CTF-based soft touch actuator, which shows high bending deformation under ultralow input voltages, be demonstrated as a soft robotic touch finger on fragile

displays. The paper itself is interesting, and it extends the novel application of CTFs. However, there are some obvious deficiencies in the synthesis and characterization of CTFs, which is the key material for the soft touch actuator.

1) The author used ionothermal process in a sealed ampoule at high temperature to synthesize CTFs, as literature reported, when temperature is 400 °C, the resulting CTFs samples show black colour, which means the high temperature has destroyed the structure and caused partial carbonization. In this paper, the authors even used higher temperature, such as 500 °C and 600 °C, how to keep CTFs integrated without carbonization?

2) When the temperature is 600 °C, how about the stability of the PIMs? Thermal stability data should be supplied.

3) For TP4, TP5, and TP6 CTFs, what reason causes different properties?

4) In the context, the authors claimed "Micropores (pore size < 2 nm) in CTFs (seen in Figure 1d-f) were not observed in Figure 2b, as this distribution is obtained from nitrogen adsorption-desorption isotherm and nitrogen has very low diffusion under ultralow relative pressure ($\sim 10^{-7}$)", however, this statement is not correct. The reason is that adsorption data at low pressure, which reflects the existence of the micropore, had not been recorded during N₂ adsorption-desorption isotherm analysis (Fig. 2a).

5) The chemical structure should be clarified further. Besides qualitative analysis, some quantitative analysis is also needed, such as the content of triazine ring.

6) In the preparation of soft ionic actuators, how to determine the dosage of CTFs? Any effects on actuators' performance?

Based on the above comments, the paper cannot be accepted in its present form.

Authors' Responses to Reviewer #1's Comments

[Reviewer #1's Overall Comment] Mahato et al. have used cyclic triazine frameworks to prepare capacitor based soft touch actuators for playing an electronic piano. The work utilizes the high surface area, heteroatom dopants of polymer of intrinsic microporosity (PIM), and thus relatively high capacitance to obtain a displacement of 17 mm under 0.5 V of square wave voltage. **The results demonstrate good use of porous materials for actuation.** The work, however, has a few fundamental problems that must be addressed.

[Answer to Reviewer #1's Overall Comment] We are greatly thankful to the reviewer for careful review of our manuscript and providing positive opinion about the use of heteroatom dopants PIM-1 based porous CTFs materials for soft touch actuators. We also thank to the reviewer for his additional comments, as those fundamental properties are really important and rational. We did our best to improve the manuscript after incorporating all comments raised by the reviewer as mentioned below.

[Comment (1)] First of all, the authors have completely miscalculated the capacitance. The authors have reported capacitance values in the range of 300-525 F/g for the devices. If the authors have double-checked the literature, they should have realized that the value of 525 F/g is an exceptionally high gravimetric capacitance, substantially higher than many other materials. After carefully checking the section of "Materials and Methods", the reviewer has found that the authors have mis-calculated the capacitance by a factor of two. In other words, equation (1) is completely wrong and the authors have missed a pre-factor of 1/2. The reviewer is afraid that the authors should not have made a mistake like this for a manuscript submitted to this journal.

[Answer to Comment (1)] We highly appreciate the reviewer's punctilious comment on calculation of specific capacitance. The followings are our clarification on calculations of specific capacitance. Generally, the specific capacitance is defined as the ratio of the stored charge per unit mass of a system, to the corresponding change in its electric potential:

$$C_{sp} = \frac{Q}{SV} \quad \text{RE1}$$

where, Q is the stored charge in coulombs, S is the mass of the active material, V is the potential, and C_{sp} is specific capacitance. Considering the current as $I = Q/t$ and the scan rate as $v = V/t$, the specific capacitance could be rewritten in terms of these parameters dividing the nominator and denominator by t :

$$C_{sp} = \frac{Q/t}{SV/t} = \frac{I}{Sv} \quad \text{RE2}$$

And so:

$$I = S \times v \times C_{sp} \tag{RE3}$$

Integrating this equation with respect to potential yields:

$$\int_{V_1}^{V_2} IdV = \int_{V_1}^{V_2} S \times v \times C_{sp} dV \tag{RE4}$$

Considering the parameters in the right hand constant, the capacitance would be calculated as:

$$C_{sp} = \frac{\int_{V_1}^{V_2} IdV}{S \times v \times \Delta V} \tag{RE5}$$

which is the equation we initially provided in supporting information without further explanation as it is widely used in electrochemistry. However, in order to obtain the specific capacitance from the CV curve, further calculations are carried out.

In typical CV curves, three area could be considered: A_1 (the area below charging curve), A_2 (the area below discharge curve), and A (the area surrounded by the charge/discharge curves). Supposing the capacitance constant during charging and discharging, the area A_1 could be written as:

$$A_1 = \int_{V_1}^{V_2} IdV = \int_{V_1}^{V_2} S \times v \times C_{sp} dV = S \times v \times C_{sp} (V_2 - V_1) \tag{RE6}$$

Similarly, the area bellow discharge curve could be written as:

$$A_2 = \int_{V_2}^{V_1} IdV = \int_{V_2}^{V_1} S \times v \times C_{sp} dV = S \times v \times C_{sp} (V_1 - V_2) \tag{RE7}$$

Now, the area A , inside the CV curve, is calculated as:

$$A = A_1 - A_2$$

$$A = S \times v \times C_{sp}(V_2 - V_1) - S \times v \times C_{sp}(V_1 - V_2)$$

$$A = S \times v \times C_{sp}(V_2 - V_1) + S \times v \times C_{sp}(V_2 - V_1)$$

$$A = 2 \times S \times v \times C_{sp}(V_2 - V_1) \tag{RE8}$$

Therefore, the specific capacitance is obtained as:

$$C_{sp} = \frac{A}{2 \times S \times v \times (V_2 - V_1)} \tag{RE9}$$

We guess the reviewer may concerns about coefficient 2 in the denominator of this equation, which is already considered in our calculations. However, we also agree with the reviewer that Equation RE5 could be confusing. Accordingly, we modified the formula in the manuscript as follows:

$$C_{sp} = \frac{A}{2 \times \Delta V \times v \times S}$$

where A is the area inside CV curve, ΔV is the potential window, v is the scan rate, and S is the weight or area taken of sample.

Regarding the high value of specific capacitance (525 F/g), authors like to draw reviewer's attentions towards some of the most recent development of high capacitance CTFs materials, reported in highly reputed journal articles.

For example:

1. Zhibin Ye *et al.* [R1] reported that CTF-800, derived from polyethynylbenzotrile, shows specific capacitance of **628 F/g** when measured under similar three-electrode system with aqueous 1M H₂SO₄ electrolyte and used it for high-energy supercapacitors.
2. S-W Kuo *et al.* [R2] reported that a Car-CTF materials, synthesized by using similar ionothermal process using [9,9'-bicarbazole]-3,3',6,6'-tetracarbonitrile as a precursor, shows specific capacitance of **545 F/g** and used it for high performance energy storage devices.
3. A. Bhaumik *et al.* [R3] reported that triazine based covalent organic frameworks, derived from the reaction of 1,3,5-tris-(4-aminophenyl)triazine and 2,6-diformyl-4-methylphenol, show specific capacitance of **418 F/g** although it has relatively low surface area of 651 m²/g. They used this material for high-performance capacitive energy storage.

Therefore, it is obvious for CTFs materials to have high value of specific capacitance. The hierarchical porosity with micro/meso/macropores as these highly porous materials

contains good percentages of heteroatom nitrogens with electronic conjugated structures, facilitating the fast ion response and charge transfer to electrode surface areas. In the reported CTFs structures, there is not only heteroatom nitrogen in the conjugated framework, but also it contains heteroatom oxygen, which has direct influence on the triazine frameworks and could be acted as energy pockets (**Figure S6a**: discussed briefly in the main manuscript) for their relatively higher specific capacitances as shown below.

Figure S6. Possible mechanism for high charge storage capacity of novel CTFs. (a) Accommodation of positive (A-B) and negative (A-C) ions in presence of dibenzo-*p*-dioxine unit.

[Comment (2)] *In the introduction, the authors commented that “porous carbon structures have some limitations in terms of charge accumulation and transfer because they lose extended electronic conjugations during carbonization. Therefore, the stored charges on electrode surfaces do not delocalize quickly during charge-discharge cycles, resulting in [a] high order of phase delay and low actuation speed.” This statement is wrong or at least highly questionable. Upon carbonization, carbon atoms are becoming more and more conjugated and eventually graphene-like or graphite-like, resulting in a fully conjugated system with high charge mobility. In fact, for many carbon materials, the capacitance becomes fully electrical double-layer and the phase delay will start to diminish. Thus, the argument that they lose conjugation and result in slow charge/discharge cannot stand (unless the authors have data to prove it or use references to back it up otherwise). The authors may have neglected that, for some porous carbon materials, the phase delay can be caused by reasons such as material mechanical properties.*

[Answer to Comment (2)] Authors completely agreed with the reviewer comments about the increase of electrical conductivity upon carbonizations due to graphite-like configurations. We modified the sentence properly by following reviewer’s comment and revised in the main manuscript as follows-

“However, porous carbon structures derived mainly from MOFs and COFs have lost their inherent chemical structures during carbonization at temperature higher than 400

°C because of thermal instability. Although some organic counter parts are converted into graphite-like short-range structures and slightly increased the electrical conductivity after carbonization, those materials are lacking of long-order π -conjugated frameworks. Therefore, theoretically the stored ionic charges on electrode surfaces do not delocalize as quickly as shown in a fully conjugated electronic long-range system, like CTFs, during charge-discharge cycles, resulting in smaller order of phase delay and relatively lower actuation speeds. In addition, the lack of extended long-order electronic conjugation can put a limit on saturated bending displacement and also can produce certain back-relaxation during long-term exposure under direct current inputs.”

Here, authors only made a comparative statement in between porous carbon (derived from most of MOFs and COFs) and CTFs macromolecules (fully π -conjugated extended cyclic triazine frameworks) without diminishing the reputation of each electrode material with respect to high actuation speed and low phase delay. Both materials have the ability for such properties, but in comparison to CTFs, porous carbon has relatively low efficiency. Because CTFs have well defined heteroatom-rich π -conjugated long-order macromolecular electronic structures that facilitate higher charge storage on the electrode surface, it helps to discharge quickly and thereby actuation speed might be increased.

[Comment (3)] *On page 8 and 9, the authors claim that “the highly porous networked structure facilitates easy ion diffusion and high electrical conductivity.....The increased concentration of electron-rich nitrogen in the triazine frame works, along with oxygen, can potentially enhance the surface charge density and directly contribute to the higher electrical conductivity and charge storage capacity”. This statement is questionable, and it needs supporting evidence. Can the authors explain why porous networked structure will facilitate electrical conductivity? It is understandable the porous network facilitates ion diffusion, but it is counterintuitive to enhance electron mobility.*

[Answer to Comment (3)] Authors are thankful to the reviewer for drawing attention on this particular sentence, “the highly porous networked structure facilitates easy ion diffusion and high electrical conductivity”. It is true that ion diffusion increases with the increase of porous structures, but there is no direct relationship to the variation of electrical conductivity on it. Here, the reported CTFs not only have highly porous structures but also they contain π -conjugated extended triazine networks with heteroatoms, nitrogen and oxygen, in the regular frameworks. Presence of those heteroatoms having dissimilar electro-negativities than regular carbon atoms can enhances the mobility of electrons as those centres have higher negative charges. So electrons are naturally forced to move towards those centres and this movement of electrons happened throughout the CTFs as they are connected by alternative single and

double bonds (long-ordered π -conjugation). This is similar to the fact that heteroatom (mainly, nitrogen) doping on the regular structures of graphene shows much higher electrical conductivity than the pristine graphene [R4]. For more clarity to the readers, authors carefully modified this portion in the main manuscript as,

“The main goal of synthesizing CTFs based on PIM-1 is to utilize the intrinsic microporosity of PIM-1, while enhancing the electrical conductivity due to the formation of electronically conjugated long-order cyclic triazine frameworks and high specific capacitance due to the generation of hierarchical porous heteroatom-containing polar surfaces. The porous structures help to increase the ionic diffusion and electronically π -conjugated long-order triazine frameworks increase the electrical conductivity. Therefore, the reported CTFs having highly porous π -conjugated network structures and heteroatom contents facilitate both of easier ion diffusion and higher electrical conductivity; the enhanced specific capacitance provides higher charge storage functionality, which is crucial for developing high-performance electro-ionic soft actuators. The increased concentration of electron-rich nitrogen in the triazine frameworks, along with oxygen, can potentially enhance the surface charge density and directly contribute to the electrical conductivity and charge storage capacity.”

[Comment (4)] *In the TGA analyses, the authors made a conclusion that the series of TP4, TP5, and TP6 have the same decomposition temperature of 670 C. However, Figure S2 shows contradictory data with completely different conclusions. The decomposition temperatures of these materials are completely different, and they increase in the sequence of TP4, TP5, and TP6. The authors should have used a scientific protocol to analyze the data properly, for example, the standard 5% weight loss.*

[Answer to Comment (4)] Authors would be happy if the reviewer draws re-attention on the TG-DTG spectra of **Figure S2a** as provided below. It is clearly observed in the DTG plot that 670 °C temperature is attributed to the decomposition of the residual part of the polymeric skeleton for TP4, TP5 and TP6 CFTs. Only the rate of decomposition is decreased from TP4 to TP6 as reflected in TG plots where the residual weight (%) of those CTFs after 800 °C temperature increases respectively. Authors believe that there was misunderstanding in between decomposition temperature and residual weight.

Figure S2. Thermogravimetric and morphological characterization. (a) TG-DTG spectra of PIM-1, TP4, TP5, and TP6. (b) Magnified SEM image of TP6.

[Comment (5)] The N₂ isotherms of TP4, TP5, and TP6 show similar pore sizes of 2.5, 3.5, and 5.1 nm. Can the authors explain or comment on why they show these particular pore sizes? In addition, rapid uptake of argon implies permanent microporosity of TP6. What do the authors mean by “permanent”?

[Answer to Comment (5)] Surface areas and pore sizes are the inherent properties of a material in solid state. These properties are straightforwardly dependent on the packing of each unit of molecule/polymer in a confined lattice structure having minimum energy, *i.e.*, thermodynamically and kinetically controlled stable state of the materials. And to make this stable configuration, there is an adjustment/rotation of the chemical bonds in three-dimensional space, generating pores in their structures in solid state. Here, all CTFs, TP4, TP5, and TP6 were obtained from the same precursor, PIM-1, *via* similar ionothermal methods with different temperatures. Therefore, all the reported CTFs have similar types of chemical functionalities and structures and show similar type of molecular interactions during packing in solid states, resulting in similar pore sizes. But different in their respective pore volumes due to the formation of different sizes of condensed aromatic triazine frameworks with different synthetic temperatures.

Here, authors are expressing the nature of reported CTFs by the term “permanent microporosity”. The term is generally used when there is an adsorption of guest molecules (here argon) in very low relative pressure (p/p^0). Because microporous materials contain pores of up to 2.0 nm in dimension and display permanent porosity due to the retention of their structural integrity in the absence of a guest [R5]. Here, all the reported CTFs were synthesized from PIM-1. This PIM-1 is a well-known material due to its unique microporous structure originated from its inherent configurations containing rigid spiro-linkages and macromolecular fuse-rings. Because of those stable spiro-linkages and fuse-rings with bonds out of benzene-ring plane, the

macromolecules have three-dimensional geometry instead of two-dimensional. The three-dimensional structure with rigid configuration prevent the PIM-1 molecules from approaching close to each other and stacking. Therefore, the materials have permanent microporosity[R6]”

Figure S6. (b) Synthetic route of microporous PIM-1 with proposed chemical structure.

[Comment (6)] *The CO₂ isotherms of TP5 and TP6 show similar uptake. Can the authors elaborate the cause of this phenomenon, especially considering their completely different surface areas?*

[Answer to Comment (6)] The CO₂-uptake capacity of a material is not only dependent on the surface area, but also the CO₂-philic interaction in between them [R7]. It has been well explored that nitrogen and hydroxyl functional groups have strong interactions with CO₂. There are many reports in the literature, which describes the phenomenon [R8-R11]. Here, although TP6 has higher surface area than TP5 by 121 m²/g, the nitrogen content of it is slightly lower than TP5 as observed in elemental analysis (Table S1). Therefore, authors believed that overall effects on CO₂-uptake capacity of TP5 and TP6 are similar by considering both of surface area and CO₂-philic nitrogen contents.

[Comment (7)] *In the Raman analysis, the authors have calculated the G and D bands ratios to be in the range of 1.47 to 1.58. Does this mean the materials have become carbon instead of remaining as polymers? In what range of G/D these materials can be still called polymers of intrinsic microporosity?*

[Answer to Comment (7)] Authors already mentioned in the manuscript that the broad D and G bands at 1354 and 1604 cm⁻¹ are similar to the fused aromatic cluster in disordered carbon structure [R12], confirming the presence of condensed triazine frameworks and benzene rings [R13] in the reported CTFs. Authors did not mentioned that the materials became carbon rather than polymer. The observed G and D bands are

very common in CTFs structures due to the formation of condensed triazine frameworks and benzene ring at high temperature. At elevated temperature, some percentages of less stable non-aromatic functional groups are decomposed and aromatic parts are condensed. Results show that increase in size of aromatic framework structures is observed in Raman analysis with higher value of I_G/I_D .

It is not possible to conclude whether the material is only carbon or remains its frameworks by only checking the G and D bands. It is very common practice to check the framework structures by FT-IR analysis, as only carbon materials do not show any characteristic peak due to absence of polar bonds. If there is no characteristic FT-IR peak corresponding to some polar chemical bonds and the materials show only G and D bands in Raman analysis, then we can say that it contains only carbon.

[Comment (8)] *It is highly recommended that authors perform mechanical property analysis of the materials used here. The mechanical properties can play an important role in the actuation and recovery of the 'fingers'. It is possible, and to some extent, it is impossible to rule out that the mechanical properties can play a more critical role than the surface area and heteroatoms that have induced the increase in capacitance.*

[Answer to Comment (8)] Authors newly performed the mechanical properties analysis of each actuators layers and reported in Supplementary Information section as **Table S2** shown below. It has been observed that PEDOT-PSS-CTF (TP6) exhibited highest tensile strength and Young's modulus than the other layers used for the fabrication of CTFs-based ionic soft actuators. This implies the strong intermolecular interaction in between PEDOT-PSS and CTFs. This observed data also supports the proposed mechanism (**Figure 3g-C**) for the electronic interaction of CTFs with PEDOT-PSS towards the increase of surface charge and electric conductivity.

Table S2. Mechanical properties of the actuator layers.

Membrane	Young's modulus, MPa	Tensile strength, MPa	Elongation at break, %
Nafion	24.10	1.36	10.89
PEDOT-PSS	36.01	3.62	35.20
PEDOT-PSS-CTF (TP6)	48.56	4.85	31.52

[Comment (9)] *Lastly, the actuation is mostly investigated at 0.1 Hz. At this frequency, it takes about 10 seconds for each actuation. Will this be relevant in practical use? Will*

someone wait for 10 seconds for each actuation? As shown in Figure 4d, at 1 Hz the peak to peak displacement becomes drastically smaller. Will this be sufficient for practical use?

[Answer to Comment (9)] Authors fully understand the importance of the raised statement by the reviewer. Authors still believe that the reported air-stable and durable soft actuator can be used for practical purposes as those actuators already showed their credibility by successfully demonstrating as a soft touch fingers array for playing electrical piano on a smart phone device. On contrary, it is not always mandatory that every practical application of soft robotics demand high displacement at 1 Hz or higher frequency, some of them may be operated by higher bending actuation at low frequencies (≤ 1 Hz) or low bending actuation at high frequencies (≥ 1 Hz). The reported actuators can be utilized for those practical applications. For other practical applications of soft robotics, demanding high actuation at very high frequencies is in progress.

Authors' Responses to Reviewer #2's Comments

[Reviewer #2's General Comment] The authors present variant of ionic EAP actuators like IPMC that are based on double layer charging. In this case it has two PEDOT-PSS electrodes doped with triazine molecules (CTF). Ions migrate between the two electrodes inducing a bending. This kind of actuators is well known and characterized in the field: classical IPMC (see works of Kim), carbon-based materials (works of Aabloo) or conducting polymers (works of Vidal). The really **interesting new novelty of this work** is that their actuators do not show the so-called back relaxation, which is a common issue with IPMC actuators.

[Answer to Reviewer #2's General Comment] Thank you so much for your comment on new novelty of this work.

[Comment (1)] The title (and focus of the paper) is misleading and should focus on the science, not an obscure demonstration. "Playing the piano" sounds much more than it actually is. The "playing the piano" part is not very impressive nor is it true. The bending actuator array only touches/taps a smart phone, which is chosen to run a piano app. This action does not take any force. All EAP actuators can do this. It is a fun demonstration from the authors, but this fits a demo-session at a conference or a public outreach event to generate interest for high school students.

[Answer to Comment (1)] The title is not "~for playing the piano", but "~ for playing electronic piano". We are not insisting that the actuator can be used for playing real piano. In this paper, we are focusing on soft touch finger which can be used to play on electronic piano app as a new functionality of electro-ionic soft actuator. The comment on "all EAP actuators can do this" is NOT true. Until now, there is no demonstration that any EAP plays electronic piano. Many actuators have reported with only basic characterizations. Bringing those actuators to further applications or demonstrations actually requires not only actuators with new materials for excellent properties, but also great effort of researchers to handle many other things rather than actuators. The reviewer mentioned about IPMC, the fact is that these IPMC actuators with metallic electrode layers can make scratch on fragile display surfaces because of rough and uneven metal surfaces. Dielectric elastomer actuators showing in-plane deformation cannot be used for this demonstration. This soft touch fingers can be used for remotely controlled systems as well as fun demonstrations, which can inspire your brain for future applications. We believe that this is an interesting result and a practical science. Most importantly, we want to make a highlight that we for the first time demonstrate the array of soft touch fingers for playing electronic piano, which demonstrates the importance of the new material we made for enhancing actuation performance, and a significant step bridging gap between current research and future practical applications.

[Comment (2)] Also, the title indicates a new type of actuator having a unique soft touch, but most (if not all) ionic actuators have soft touch. This is actually their biggest drawback over electronic DEAs. The lack of useful (blocking) force. If the authors had developed a soft actuator that could play a real (mechanical) piano, i.e. an EAP actuator that has the stroke and force to press a real piano key, that would have a nice breakthrough worth this journal.

[Answer to Comment (2)] Playing real piano is possible when we use conventional robotic actuators. That is NOT a breakthrough in robotic systems. There are so many powerful actuators. In this paper we are not saying that the developed actuator is almighty actuator. We are reporting that we newly synthesize new CTF electrode materials suitable for electro-ionic soft actuators and show very interesting demonstration for playing electronic piano using electro-ionic soft touch finger. We added blocking force in the revised manuscript. The blocking force is over 20 times compared with weight of the actuators.

[Comment (3)] The authors should revise their manuscript considerably. Focusing on the science and downplaying or removing the piano part. Also, the text should be written in a more scientific way and not as a sales brochure. The text is full of sales pitch words e.g. “exceptionally”, “excellent”, "superlative”, etc. Interesting results but they fit a more targeted, narrower audience.

[Answer to Comment (3)] We revised again in a more scientific way. Thank you for your comments.

Authors' Responses to Reviewer #3's Comments

[Reviewer #3's Overall Comment] *This paper is concerned about a new electrode material of electroactive polymer (EAP) actuator. In this manuscript, the authors successfully fabricated flexible electrodes for ionic EAP actuators by preparing the composites of PEDOT/PSS with synthesized PIM-1 based CTFs which have a porous structure and chemical affinity with ions. Furthermore, they showed that the fabricated actuators had an excellent performance of large displacement and explored a basic mechanism of its performance by analyzing the structure of electrodes perfectly. Hence, I think that **this paper will be publishable in Nature Comm.** Before publishing, however, if the following points would be revised, I think that this paper must be more useful for the readers in related research fields. So, the authors should be considered and revised the following points:*

[Response to Reviewer #3's Overall Comment] Authors are very grateful to the reviewer for the positive comments about the impact of this novel research finding and its usefulness towards the related research communities. Authors are happy to revise the review's mentioned comments point-by-point to improve its significance in broader scale.

[Comment (1)] *Page 6, line 10: A word, "a polymer of intrinsic microporosity (PIM-1)" is an important key word that shows polymer material. Here, the authors should define what it is by using Figure S6(b) exactly.*

[Answer to Comment (1)] Authors define PIM-1 by considering its structure (**Figure S6b**) and we revised the following sentence in the revised manuscript.

"The polymer of intrinsic microporosity (PIM-1) pioneered by Peter Budd and Neil McKeown³⁵ be a perfect precursor for the synthesis of novel CTFs having both of porosity and surface charge capacitance. PIM-1 is a well-known material due to its unique microporous structure originated from inherent rigid spiro-linkage and macromolecular fuse-ring configurations (Figure S6b) that can resist to pack efficiently in solid state³⁵, resulting in permanent microporosity on it³⁵: it has both nitrogen and oxygen heteroatoms on its backbone and can easily soluble in common organic solvents like CHCl₃, THF, etc."

Figure S6. (b) Synthetic route of microporous PIM-1 with proposed chemical structure.

[Comment (2)] In the manuscript, the performances of the actuator are evaluated by the displacements. But the displacement is known to depend on the dimension of the actuator film. According to Table S3 that summarizing “Comparison of bending performance of ionic soft actuators”, there are no data of the thickness of each actuator film. It is well-known that the bending displacement increases with decreasing the thickness of the actuator. So, the authors should calculate the dimensionless parameters from the displacements, and discuss the performances of the fabricated actuators.

[Answer to Comment (2)] Authors are thankful to the reviewer for his useful comment about the comparative study of actuation performances by using **thickness** data. **Table S3** (shown below) is accordingly revised and thickness of mentioned ionic soft actuators are added freshly as a separate column to justify the comparison of bending actuation performances of the TP6PP actuator with previously reported ionic soft actuators. From **Table S3**, it is observed that most of the reported bending displacements of ionic soft actuators is measured by taking the thickness in the range of 80-115 μm . The TP6PP ionic soft actuator with thickness of 115 μm shows much larger bending deflection at relatively low input voltage of 0.5 V. Authors carefully revised the main manuscript (page: 18-19) by adding this important parameter.

Table S3. Comparison of bending performance of ionic soft actuators.

Ionic soft actuator	Input potential (sine wave) & frequency	Length, mm	Thickness, μm	Bending displacement (peak-to-peak), mm	Ref.
3D G-CNT-Ni/PP	± 1.0 V and 0.1 Hz	24	248	4.84	18

PP/3D GCN-NG	± 0.5 V and 0.1 Hz	18	100	6.50	21
GM-NG	± 3.0 V and 0.1 Hz	-	-	6.40	23
BS-COF-C900/PP	± 0.5 V and 0.1 Hz	20	85	8.60	25
Th-SNG/PP	± 0.5 V and 0.1 Hz	26	90	4.60	26
HPNC-900/PP	± 0.5 V and 0.1 Hz	20	100	6.99	27
p MoS ₂ - n SNrGO nanohybrid	± 0.5 V and 0.1 Hz	20	105	9.90	46
HLrGOP	± 5.0 V and 0.1 Hz	20	-	8.52	47
PS- b -PSS- EMIm/EMImBF ₄ electrolyte based NS co- doped graphene/PP electrodes	± 0.5 V and 0.1 Hz	20	80	6.80	48
IL-IPMC	± 10.0 V and 0.1 Hz	30	313	11.10	49
Nacre-based carbon nanomeshes electrode	± 3.0 V (Sq.) and 0.1 Hz	30	165	10.00	50
BP-CNTs/CNTs	± 1.5 V (Sq.), 0.1 Hz	23	115	12.50	7
Nafion and Pt based IPMC	± 1.0 V, 0.1 Hz	25	-	2.00	51
TP6PP	± 0.5 V and 0.1 Hz	25	115	13.50	This work

The width of the fabricated ionic soft actuators used in this research article is 3.0 ± 0.017 mm.

PP: PEDOT-PSS

Regarding presentation of results in terms of bending strain (as non-dimensional parameter) we would prefer to avoid as it is a controversial issue these days. Most of the papers in the literature, who reported the strain, used the following equation to

calculate the strain of the actuator, which is not accurate and somehow mislead the calculations:

$$\varepsilon = \frac{2t\delta}{L^2 + \delta^2} \tag{RE10}$$

This equation was initially introduced to refer the strain difference between upper and lower surfaces of the actuator taking some simplifications [R14]. However, many scholars have used this formula to report the maximum strain generated by their actuators. To accurately calculate the maximum bending strain, the following methodology should be considered in the linear range.

When a beam is subjected to pure bending it would deform into a uniform arc. In such case, the upper face of the beam goes through contraction and the lower face experiences expansion. Accordingly, there must exist a surface parallel to the upper and lower faces of the member, where the length remain unchanged. This surface is called the *neutral surface* and places at the middle of the beam cross-section for those with symmetric cross-sections like rectangular or circular shapes [R15]. The origin of coordinates is usually selected on the neutral surface, rather than on the lower face of the member as done earlier, so that the distance from any point to the neutral surface will be measured by its coordinate y .

In the above figure ρ represent the radius of arc DE and θ is the central angle corresponding to DE , while the length of DE is equal to the length L of the undeformed member. Therefore:

$$L = \rho\theta \tag{RE11}$$

Considering the arc JK located at a distance y above the neutral surface, the length of this arc, L' , could be calculated as:

$$L' = (\rho - y)\theta \tag{RE12}$$

Since the original length of arc JK was equal to L , the deformation of JK is

$$\Delta L = L' - L = (\rho - y)\theta - \rho\theta$$

$$\Delta L = -y\theta \tag{RE13}$$

The longitudinal strain, ϵ_x , in the elements of JK is obtained by dividing ΔL by the original length L of JK .

$$\epsilon_x = \frac{\Delta L}{L} = \frac{-y\theta}{\rho\theta}$$

$$\epsilon_x = -\frac{y}{\rho} \tag{RE14}$$

It is clear that the maximum strain, ϵ_m , occurs at beam upper or lower surfaces. The position of upper and lower surfaces with respect to neutral surface would be $y = \pm \frac{t}{2} = \pm c$, for the rectangular cross-sections (t denotes the thickness of the beam).

Therefore, the maximum strain could be written as:

$$\epsilon_m = \pm \frac{c}{\rho} \tag{RE15}$$

Therefore, we can write

$$\epsilon_x = -\frac{y}{c} \epsilon_m \tag{RE16}$$

Considering the Hooke's law, $\sigma_x = E\epsilon_x$, similar equation could be written for the stress at cross-section of the beam:

$$\sigma_x = -\frac{y}{c} \sigma_m \tag{RE17}$$

The moment M could be written in terms of stress by the moment of equivalent forces applying on the cross-section:

$$\int_{-c}^c y\sigma_x dA = -M \tag{RE18}$$

Substituting σ_x :

$$\int_{-c}^c y\left(-\frac{y}{c}\sigma_m\right) dA = -M \tag{RE19}$$

Or

$$\frac{\sigma_m}{c} \int_{-c}^c y^2 dA = M \tag{RE20}$$

In above equation the integral $\int_{-c}^c y^2 dA$ is equal to second moment of inertia of the cross-section, I . Hence, the maximum stress could be written in terms of moment as follows:

$$\sigma_m = \frac{cM}{I} \quad \text{RE21}$$

And therefore,

$$\sigma_x = -\frac{yM}{I} \quad \text{RE22}$$

Substituting the strain from Hook's law into above equation yields:

$$\varepsilon_m = \pm \frac{cM}{EI} \quad \text{RE23}$$

On the other hand, we know that the tip displacement of cantilever beam could be calculated as follows for small deflections [R16]:

$$\delta = \frac{ML^2}{2EI} \quad \text{RE24}$$

Where, δ is the tip displacement of the beam. Calculating the moment(M) in term of tip displacement (δ) and substituting in the strain equation yields:

$$\varepsilon_m = \pm \frac{2c\delta}{L^2} \quad \text{RE25}$$

Since the thickness of the beam is equal to $2c$, the maximum strain would be calculated as:

$$\varepsilon_m = \pm \frac{t\delta}{L^2} \quad \text{RE26}$$

Using this equation yields much lower bending strain (almost half) than those calculated by equation RE10. Therefore, it is not likely to make a fair comparison with other reported actuators if we accurately report the strain using equation RE26. Accordingly, we would prefer to keep our data in the manuscript based on bending displacement with thickness and length data. *However, by following the reviewer's request, we calculated the bending strain (%) of the reported CTFs-based ionic soft actuators using the equation RE26 at ± 0.5 and 1.0 V square wave input potentials under excitation frequency of 0.1 Hz as shown below.* TP6PP shows highest bending strain of ± 0.5 % based on RE26 (*equivalent to bending strain difference of 1.0 % based on RE10*) in 1.0 V square wave compared with other soft ionic actuators. In comparison to pristine PP actuators, TP6PP soft actuator showed more than three times higher bending strains.

Figure R1. Typical actuating performances of fabricated PP, TP4PP, TP5PP, and TP6PP ionic soft actuators. Time-dependent bending strain (%) of actuators under (a) ± 0.5 V and (b) 1.0 V square wave input potential under 0.1 Hz excitation frequency.

[Comment (3)] *In the manuscript, the authors discuss fast transfer of ions in the electrode in relation to its porous structure. In order to discuss the ionic and electronic transfer of electrodes, the authors should add the electrochemical impedance measurements of fabricated actuators, and discuss the impedance data in the manuscript or the supplementary information.*

[Answer to Comment (3)] Authors are thankful to the reviewer for mentioning the electrochemical impedance, which is related to the fast transfer of ions in the electrode. Authors newly did experiments on the electrochemical impedance measurements and added **Figure S8** as shown in in the revised version of Supplementary Information.

A Nyquist plot of all CTFs-based actuators show a semi-circle at high frequency region related to charge transfer resistance (R_{ct}) and a straight-line at low frequency region related to solid-state diffusion of ions. It has been observed from the spectra that TP6PP actuator shows smallest R_{ct} value (28 ohms) than those of the TP5PP (39 ohms) and TP4PP (63 ohms) actuators. These results also support the higher order of charge transport in the TP6PP actuator in comparison to others and correspondingly TP6PP shows relatively high areal capacitance.

Figure S8. Electrochemical impedance spectra (EIS) of TP4PP, TP5PP, and TP6PP ionic soft actuators. Inset: The magnified high frequency region.

[Comment (4)] *The data of the generated force summarized in Table S4 seems to be not good or usual as compared to that of other ionic polymer actuators previously reported. The authors should discuss the reason for this and how to improve this based on the proposed electrode material in this manuscript.*

[Answer to Comment (4)] Authors are thankful to the reviewer for his valuable suggestion regarding the further increment of blocking forces of the reported actuators. Authors are happy to revise the manuscript by describing the obtained blocking force (Table S4) for CTFs based electrode materials and the possible ways to improve it further as suggested by the reviewer below. The maximum blocking force of the TP6PP actuator under DC input was also measured with various amplitudes of the stimulation. As could be seen in Table S4, the developed actuator is able to generate relatively high blocking force, 24 times of its weight, under 2.0 V stimulation. Comparing this data with those of PP actuator reveals that the force generating capability of the actuator is improved up to almost 100% by simply adding CTFs to the PP electrode. It is very promising result for a metal free ionic soft actuator where a light-weight fully organic cyclic triazine frameworks is used for active electrode materials. In addition, there is tremendous opportunities to increase the blocking forces further in near future by simply increasing the thickness of the actuators or utilizing stiffer ionic polymer membranes rather than Nafion. Besides, there is a possibility to incorporate some heavy conducting metals inside the pores of it or grow some metal nanowire on CTFs, which not only prevent the air-oxidation of metal but also increase the conductivity and specific capacitance of the electrodes in many folds.

Authors' Responses to Reviewer #4's Comments

[Reviewer's Overall Comment] *In this paper, Prof. Oh reported a CTF-based soft touch actuator, which shows high bending deformation under ultralow input voltages, be demonstrated as a soft robotic touch finger on fragile displays. The paper itself is interesting, and it extend the novel application of CTFs. However, there are some obvious deficiencies in the synthesis and characterization of CTFs, which is the key material for the soft touch actuator.*

[Response to Reviewer 4's Overall Comment] Authors are greatly thankful to the reviewer for his appreciation of this novel application of CTFs as a soft robotic touch finger on fragile devices and his valuable comments on CTFs which is playing the key role for this whole finding. All those raised comments are made clear at our best one by one as follows for the kind consideration.

[Comment (1)] *The author used ionothermal process in a sealed ampoule at high temperature to synthesis CTFs, as literature reported, when temperature is 400 °C, the resulting CTFs samples show black colour, which means the high temperature have destroyed the structure and cause partial carbonization. In this paper, the authors even used higher temperature, such as 500 °C and 600 °C, how to keep CTFs integrated without carbonization?*

[Answer to Comment (1)] Authors completely agree with the raised comment on partial carbonization during synthesis of materials at elevated temperature. In this case, during synthesis of CTFs, authors encountered with it which is mentioned in the manuscript as disordered carbon structures. The condensed aromatic/triazine rings are clearly seen in the deconvoluted XPS peaks (**Figure 2g-i**). Those figures also confirm that with increase in temperature from 400 to 600 °C, the $N_{\text{graphitic}}$ peak area corresponds to the partial carbonization and condensed carbon structures are also increased. This fact is additionally supported by the increased ratio of G and D bands with the increase in temperature from 400 to 600 °C as shown in **Inset of Figure 2e**. *However, as the synthesis of CTFs is performed in a vacuum (free of any gas) sealed glass ampoule, the probability of decomposition by oxidative reactions is significantly reduced and the materials are able to keep their basic structures. Also extended triazine rings showed good thermal stability at high temperature.*

[Comment (2)] *When the temperature is 600 °C, how about the stability of the PIMs? Thermal stability dada should be supplied.*

[Answer to Comment (2)] Authors like to draw reviewer attention on **Figure S2a** in Supplementary Information where the thermal stability data of PIM-1 along with CTFs are reported as TG-DTG plots. From the figure as shown below, it is clearly observed that PIM-1 is stable up to 518 °C and then degradation begins to start. At a temperature of 600 °C, around 62-63% of residual weight was maintained. Still PIM-1 has much higher thermal stability than the other precursors used in the synthesis of CTFs in literature. For example, 1,4-dicyanobenzene, which is used for CTFs synthesis, has the melting point of 221-225 °C [R17-R19].

Figure S2. Thermogravimetric and morphological characterization. (a) TG-DTG spectra of PIM-1, TP4, TP5, and TP6. **(b)** Magnified SEM image of TP6.

[Comment (3)] *For TP4, TP5, and TP6 CTFs, what reason to cause different properties?*

[Answer to Comment (3)] Authors are delighted to notify about the reason to have different properties in the reported CTFs. TP4, TP5, and TP6 CTFs are ionothermally synthesized at three different temperatures of 400, 500, and 600 °C respectively. And it is reported that with increase in the synthetic temperature of CTFs, size of condensed triazine aromatic structure is also increased (characterized by Raman spectra with increased ratio of I_G and I_D ; inset of Figure 2e represents the same for the reported CTFs), resulting in a variety of molecular rearrangements during packing in solid state [R20]. Therefore, the obtained CTFs, TP4, TP5, and TP6 show different surface properties including specific surface area (Figure 2a), pore volume (Figure 2b) and heteroatoms content (Table S1).

[Comment (4)] *In the context, the authors claimed “Micropores (pore size < 2 nm) in CTFs (seen in Figure 1d-f) were not observed in Figure 2b, as this distribution is obtained from nitrogen adsorption-desorption isotherm and nitrogen has very low diffusion under ultralow relative pressure (~ 10⁻⁷)”, however, this statement is not correct. The reason is that adsorption data at low pressure, which reflects the existence of the micropore, had not been recorded during N₂ adsorption-desorption isotherm analysis (Fig. 2a).*

[Answer to Comment (4)] Authors are very much thankful to the reviewer for correcting the statement. Authors completely revised this statement in the manuscript as suggested by the reviewer.

“Micropores (pore size < 2 nm) present in these CTFs (as shown in **Figure 1d-f**) were not observed in **Figure 2b**, because the pore-size distribution curve is plotted from corresponding nitrogen adsorption-desorption isotherm data of those CTFs. And, it is obvious that adsorption-desorption data were not recorded at very low relative pressure defining micropores during N₂-sorption analysis.”

[Comment (5)] *The chemical structure should be clarified further, Besides qualitative analysis, some quantitative analysis is also needed, such as the content of triazine ring,*

[Answer to Comment (5)] Authors fully agreed with the reviewer’s comments on quantitative analysis of CTFs. Here, authors newly performed solid-state ¹³C CP-MAS NMR spectroscopic analysis as additional work for the characterization of CTFs and incorporated in the revised version of Supplementary Information section as **Figure S7** shown below.

The signal at 161 ppm is assigned to triazine ring carbon atoms [R21], which confirms the formation of triazine ring in the CTFs frameworks, while the signals at 149 and 130 ppm correspond to the phenyl ring carbons. The signal at 118 ppm is –C–O that is connected with aromatic ring. The signal at 113 ppm is characteristic for nitrile groups as well as the carbon of the phenyl ring to which the nitrile groups are bound. The significant alkyl protons appear as broad peaks at 23-57 ppm region due to high temperature synthetic procedure. The content of triazine rings in the reported CTFs is already ascertained in the manuscript by XPS (Figure 2g-i) and FT-IR (Figure 2d) spectroscopic analysis. However, it is not easy to analyze the exact numbers of triazine/benzene rings/other chemical counterparts quantitatively in CTFs as it has a long-range polymeric framework structures and difficult to get perfect crystalline peaks in powder XRD. It is interesting to characterize CTFs quantitatively and authors are working on it.

Figure S7. Solid-state ^{13}C CP-MAS NMR spectrum of CTFs (TP4).

[Comment (6)] *In the preparation of soft ionic actuators, how to determine the dosage of CTFs? Any effects on actuators' performance?*

[Answer to Comment (6)] Authors are delighted to disclose the reviewer's concern and added it in the revised manuscript under Materials and Methods section. It is always important to optimize the fabrication process before going for some applications. Here, the optimized dose of CTFs (0.35 mg/mL) with respect to PEDOT-PSS is determined

by the homogeneous dispersion of CTFs on PEDOT-PSS solution before coating on to ionic liquid-loaded Nafion surfaces and respective actuation data.

References in Responses to Reviewers' Comments

[R1] Vadiyar, M.M., Liu, X. and Ye, Z. Macromolecular Polyethynylbenzotrile Precursor-Based Porous Covalent Triazine Frameworks for Superior High-Rate High-Energy Supercapacitors. *ACS Appl. Mater. Interfaces* **11**, 45805-45817 (2019).

[R2] Mohamed, M.G., EL-Mahdy Ahmed F.M., Ahmed, Mahmoud M.M. and Kuo, S.W. Direct Synthesis of Microporous Bicarbazole-Based Covalent Triazine Frameworks for High-Performance Energy Storage and Carbon Dioxide Uptake. *ChemPlusChem* **84**, 1767-1774 (2019).

[R3] Bhanja, P., Bhunia, K., Das, S.K., Pradhan, D., Kimura, R., Hijikata, Y., Irle, S. and Bhaumik, A. A New Triazine-Based Covalent Organic Framework for High-Performance Capacitive Energy Storage. *ChemSusChem* **10**, 921-92 (2017).

[R4] Lin, L., Li, J., Yuan, Q., Li, Q., Zhang, J., Sun, L., Rui, D., Chen, Z., Jia, K., Wang, M. and Zhang, Y. Nitrogen cluster doping for high-mobility/conductivity graphene films with millimeter-sized domains. *Sci. Adv.* **5**, eaaw8337 (2019).

[R5] Bradshaw, D., Prior, T.J., Cussen, E.J., Claridge, J.B. and Rosseinsky, M.J. Permanent microporosity and enantioselective sorption in a chiral open framework. *J. Am. Chem. Soc.* **126**, 6106-6114 (2004).

[R6] Budd, P. M., Elabas, E. S., Ghanem, B. S., Makhseed, S., McKeown, N. B., Msayib, K. J., Tattershall, C. E. and Wang, D. Solution-processed, organophilic membrane derived from a polymer of intrinsic microporosity. *Adv. Mater.* **16**, 456-459 (2004).

[R7] Patel, H. A. and Yavuz, C. T. Noninvasive functionalization of polymers of intrinsic microporosity for enhanced CO₂ capture. *Chem. Comm.* **48**, 9989-9991 (2012).

[R8] Schwab, M. G., Fassbender, B., Spiess, H. W., Thomas, A., Feng, X. L. and Mullen, K. Catalyst-free preparation of melamine-based microporous polymer networks through Schiff base chemistry. *J. Am. Chem. Soc.* **131**, 7216-7217 (2009).

[R9] Dawson, R., Adams, D.J. and Cooper, A.I., Chemical tuning of CO₂ sorption in robust nanoporous organic polymers. *Chem. Sci.* **2**, 1173-1177 (2011).

[R10] Farha OK, Spokoyny AM, Hauser BG, Bae YS, Brown SE, Snurr RQ, Mirkin CA, Hupp JT. Synthesis, properties, and gas separation studies of a robust diimide-based microporous organic polymer. *Chem. Mater.* **21**, 3033-3035 (2009).

[R11] Lu, W., Sculley, J.P., Yuan, D., Krishna, R., Wei, Z. and Zhou, H.C., 2012. Polyamine-tethered porous polymer networks for carbon dioxide capture from flue gas.

Angew. Chem., Int. Ed. **51**, 7480–7484 (2012).

[R12] Ferrari, A. C. and Robertson, J. Interpretation of Raman spectra of disordered and amorphous carbon. Phys. Rev. B **61**, 14095-14107, (2000).

[R13] Li, Y., Zheng, S., Liu, X., Li, P., Sun, L., Yang, R., Wang, S., Wu, Z. S., Bao, X. and Deng, W. Q. Conductive Microporous Covalent Triazine-Based Framework for High-Performance Electrochemical Capacitive Energy Storage. Angew. Chem. Int. Ed. **130**, 8124-8128 (2018).

[R14] Takeuchi, I., Asaka, K., Kiyohara, K., Sugino, T., Terasawa, N., Mukai, K., Fukushima, T. and Aida, T. Electromechanical behavior of fully plastic actuators based on bucky gel containing various internal ionic liquids. Electrochim. Acta, **54**, 1762-1768, (2009).

[R15] Beer, F., Johnston, R., DeWolf, J.T. and Mazurek, D.F. Mechanics of Materials. 6th edition, New York: McGraw-Hill, (2012).

[R16] Young, W. C., Budynas, R. G. and Sadegh, A. M. Roark's formulas for stress and strain, 7th edition, New York: McGraw-Hill, (2002).

[R17] Kuhn, P., Forget, A., Su, D., Thomas, A. and Antonietti, M. From microporous regular frameworks to mesoporous materials with ultrahigh surface area: dynamic reorganization of porous polymer networks. J. Am. Chem. Soc. **130** 13333-13337 (2008).

[R18] Kuhn, P., Antonietti, M. and Thomas, A. Porous, covalent triazine-based frameworks prepared by ionothermal synthesis. Angew. Chem. Int. Ed. **47**, 3450-3453 (2008).

[R19] Bhunia, A., Dey, S., Bous, M., Zhang, C., von Rybinski, W. and Janiak, C. High adsorptive properties of covalent triazine-based frameworks (CTFs) for surfactants from aqueous solution. Chem. Comm. **51**, 484-486 (2015).

[R20] Li, Y., Zheng, S., Liu, X., Li, P., Sun, L., Yang, R., Wang, S., Wu, Z.S., Bao, X. and Deng, W.Q. Conductive Microporous Covalent Triazine-Based Framework for High-Performance Electrochemical Capacitive Energy Storage. Angew. Chem. Int. Ed. **57**, 7992-7996 (2018).

[R21] Dey, S., Bhunia, A., Esquivel, D. and Janiak, C. Covalent triazine-based frameworks (CTFs) from triptycene and fluorene motifs for CO₂ adsorption. J. Mater. Chem. A, **4**, 6259-6263 (2016).

At this stage, we would like to mention that the reviewer's comments were so highly pertinent and helpful that we can enhance the quality of the article and deepen the overall level of our scientific understanding. We enormously appreciate the reviewers for their careful reviews, and we hope that with such significant improvement the paper is now acceptable for publication. All of the changes in the manuscript are marked in RED.

REVIEWER COMMENTS

Reviewer #1 (Remarks to the Author):

Mahato et al. have revised the manuscript to an acceptable level. The reviewer is about to recommend the manuscript for publication, given that the authors further clarify the following points.

1. Reviewers #1 and #4 share the same concern about polymer stability at elevated temperatures. The DTA peaks at 670 Celsius degree don't mean that these polymers are stable up to 670 degrees. On the contrary, they are highly unstable at this temperature and the decomposition starts way before this temperature, probably 500 degrees based on the TGA curves. In addition, TGA typically has a fast ramp rate, and there will be a time lag for the polymers to respond to the temperature change. Thus, the polymers are usually decomposed at a lower temperature than the TGA curves show. Lastly, the vacuum environment in a sealed ampoule doesn't mean that the polymer cannot decompose. It only suggests that oxidation of the polymers is absent. The carbonization process can happen regardless of the environment. The points are as follows. Can the authors provide more evidence that there are still polymers remain in the product material, such as proton NMR? If one can see the presence of protons in the molecules, one will be more confident the polymers are only partially carbonized. The reviewer also suggests the authors discuss the above points in the manuscript.

2. In the capacitance calculation, the authors calculate the capacitance by integrating current across the voltage range. This can be problematic because the current crosses zero. A simple integration will result in a wrong area. I strongly suggest the authors calculate the area by integrating the current difference over the voltage range and divide it by two (because there are two separate processes of charging and discharging). The reviewer is sorry for being persistent on this matter. Because there are too many wrongly reported numbers in the literature, I hope the authors agree to double-check this matter to keep the community in check.

Reviewer #4 (Remarks to the Author):

The authors have addressed my concerns partially. However, the author didn't fully answer one of my questions: "For TP4, TP5, and TP6 CTFs, what reason to cause different properties?" As the authors mentioned, "partial carbonization during the synthesis of materials at elevated temperature", then how to avoid the influence of different carbon content caused by high temperature on performance? In the manuscript, the authors have attributed to the different structures of CTFs caused by different temperatures totally. The authors should give more experimental data to eliminate the contents of carbon.

Authors' Responses to Reviewer #1

[Reviewer #1's Overall Comment] *Mahato et al. have revised the manuscript to an acceptable level. The reviewer is about to recommend the manuscript for publication, given that the authors further clarify the following points.*

[Answer to Reviewer #1's Overall Comment] Authors are greatly thankful to the reviewer for his valuable and positive comments on this research content along with the recommendation for publishing in *Nature Communication*. Authors are delighted to clarify the reviewer's additional points to improve its significance.

[Reviewer #1's Comment (1)] *Reviewers #1 and #4 share the same concern about polymer stability at elevated temperatures. The DTG peaks at 670 degree Celsius don't mean that these polymers are stable up to 670 degrees. On the contrary, they are highly unstable at this temperature and the decomposition starts way before this temperature, probably 500 degrees based on the TGA curves. In addition, TGA typically has a fast ramp rate, and there will be a time lag for the polymers to respond to the temperature change. Thus, the polymers are usually decomposed at a lower temperature than the TGA curves show. Lastly, the vacuum environment in a sealed ampoule doesn't mean that the polymer cannot decompose. It only suggests that oxidation of the polymers is absent. The carbonization process can happen regardless of the environment. The points are as follows. Can the authors provide more evidence that there are still polymers remain in the product material, such as proton NMR? If one can see the presence of protons in the molecules, one will be more confident the polymers are only partially carbonized. The reviewer also suggests the authors discuss the above points in the manuscript.*

[Answer to Reviewer #1's Comment (1)] Authors are thankful to the reviewer for his well-mentioned scientific remarks on the stability of CTFs at elevated temperature. Authors strongly agreed about the usual carbonization and graphitization of reported CTFs during ionothermal synthetic environments, which are quite obvious from their $N_{\text{graphitic}}$ XPS peak as represented in the main manuscript (**Figure 3g-i**), and also shown below for your kind reference. It is also perceived that the extent of graphitization increases with the increase in synthetic temperature as the relative area of the peak increases accordingly from TP4 to TP6, which is common in ionothermal synthesis of CTFs polymers. It is true that the graphitization process starts before the mentioned temperature of 670 °C from the analysis of TGA plot, but the thermal

degradation of main backbone frameworks is initiated only after 600 °C for the TP6 polymer. Because the major loss of residual polymer weight is only observed above this temperature for TP6 (only CTFs which is synthesized at highest temperature of 600 °C). TP5 and TP4 CTFs are relatively more stable due to their lower synthetic temperatures (500 °C for TP5 and 400 °C for TP4) and correspondingly they show smaller extent of carbonization than TP6. Further to confirm the decomposition of framework structures above 600 °C, authors additionally perform the synthesis of CTFs at 700 °C (TP7) by following same experimental procedure. The obtained product (TP7) after purification was then subjected to nitrogen-sorption isotherm analysis. It was observed that surface area of TP7 is reduced noticeably to 404 m²/g from 1192 m²/g (TP6) indicating a major degradation of intrinsic porous back-bone frameworks of it (**Supplementary Figure 2b**).

Figure 3. Physicochemical and structural characterizations of as-synthesized novel CTFs electrode materials. (g-i) XPS N1s spectra of (g) TP4, (h) TP5, (i) TP6 materials. Display three N configurations after deconvoluted as olive: N_{triazine}, blue: N_{cyano}, and violet: N_{graphitic}. Increased in N_{graphitic} peak area from TP4 to TP6 reveals frameworks containing graphitic nitrogen increases with the increase in synthetic temperature.

Supplementary Figure 2b. Nitrogen adsorption-desorption isotherms of TP6 and

TP7 at 77 K. Filled and empty symbols denote adsorption and desorption, respectively. Inset: respective specific surface area data. All the measurements were carried out twice using 200 mg of each CTFs as mentioned.

Authors completely agree with the reviewer's concern about the partial carbonization of reported CTFs and thankful for your essential suggestion to characterize the presence of proton in the final CTFs structures. It is evident that the presence of hydrogen in the reported CTFs can eventually prove the retaining of framework structures on it. Therefore, according to your suggestion, authors performed solid-state proton (^1H) CP-MAS NMR spectroscopic analysis of PIM-1, TP4, TP5, and TP6 CTFs and the obtained results are shown in **Supplementary Figure 6**.

Supplementary Figure 6. Solid-state ^1H CP-MAS NMR spectroscopic analysis of PIM-1, TP4, TP5, and TP6 CTFs.

At first, solid-state ^1H CP-MAS NMR of PIM-1 (the precursor polymer of reported CTFs) is examined to understand the chemical environment of protons in the

framework structures. The three distinct peaks for PIM-1 as obtained in terms of chemical shifts (δ) are assigned to aromatic protons (\blacktriangle 6.40 ppm, deshielded), $-\text{CH}_2$ protons (\bullet 3.80 ppm, less shielded) and $-\text{CH}_3$ protons (\blackstar 0.96 ppm, shielded). In addition, the spectrum consists of broad unwanted signals, which are commonly referred as spinning sidebands (defined with \blackstar asterisks) and usually originated from the background signals during solid-state NMR analysis. Presence of those assigned protons in the framework structures are also reflected in the solid-state proton (^1H) CP-MAS NMR spectra of TP4, TP5, and TP6 CTFs. However, due to change in chemical and physical environments in CTF structures, those hydrogen binding appears at relatively lower δ values (more shielded) of 5.23, 3.33, and 0.88 ppm corresponding to aromatic protons, $-\text{CH}_2$ protons, and $-\text{CH}_3$ protons, respectively. Strong intense NMR signal of TP4, TP5, and TP6 at 3.33 ppm for $-\text{CH}_2$ protons reveals that the structural integrity of CTFs. As expected, the intensity of NMR signal for aromatic protons at 5.23 ppm decreases from TP4 to TP6 due to increase in the degree of carbonization with the increase of synthetic temperatures from 400 to 600 $^\circ\text{C}$. The $-\text{CH}_3$ peak show broad NMR signal at 0.88 ppm in all of the reported CTF polymers. In conclusion, the interpreted NMR data clearly prove the presence of hydrogen (proton) in the CTF structures in spite of their high synthetic temperatures, elucidating the retention of frameworks although there is a partial carbonization.

Authors are delighted to mention that the presence of hydrogen bonds on CTFs as $-\text{C-H}$ moiety is readily recognized from the already reported FT-IR spectra (**Figure 3d**) in the main manuscript as shown below (authors freshly marked the $-\text{C-H}$ FT-IR peak regions). Transmittance peaks as observed in between 2800-3000 cm^{-1} are related to stretching vibrations of $-\text{C-H}$ functional moiety whereas the obtained peak at 870 cm^{-1} is corresponding to out-of-plane bending vibrations of it as present in all CTFs materials including the precursor PIM-1. To clearly ascertain those FT-IR peaks corresponding to asymmetric and symmetric stretching vibration of $-\text{C-H}$ functional moiety, authors additionally provided another FT-IR plot (enlarging the $-\text{C-H}$ bond vibrational frequency region from 3,100 to 2,600 cm^{-1}) in the supporting information section (**Supplementary Figure 5**) showing clear evidences about the presence of $-\text{C-H}$ moiety on the final CTFs structures. The obtained characteristic FT-IR peaks at 2,970 cm^{-1} , 2,920 cm^{-1} , and 2,850 cm^{-1} are mainly due to the as present $=\text{C-H}$ (sp^2) stretching, $-\text{C-H}$ (sp^3) asymmetric and symmetric stretching vibrations respectively on the reported CTFs polymers.

Figure 3. (d) FT-IR spectra of PIM-1 and corresponding novel PIM-1 based CTFs. Inset: disappearance of nitrile ($\text{-C}\equiv\text{N}$) peak of PIM-1 due to trimerization and appearance of -C=N- peaks due to formation of cyclic triazine frameworks (CTFs: TP4, TP5, and TP6).

Supplementary Figure 5. FT-IR spectra of PIM-1 and corresponding novel PIM-1 based CTFs to show the presence of C-H moiety.

The integrity of framework structures of TP4, TP5, and TP6 CTFs is also reflected in the solid-state ^{13}C CP-MAS NMR spectrum (**Supplementary Figure 7**, shown below). There are mainly six different types of chemically non-equivalent carbons present in the reported CTFs, which are highlighted by spherical symbols with individual colours (**Supplementary Figure 7b: Inset**). The obtained NMR signal at 161 ppm confirms the presence of triazine ring carbon in the reported CTFs,¹ while the signals in between 129-149 ppm are corresponding to the phenyl ring carbons of as present modified PIM-1 structures. The NMR signal at 118 ppm is related to the -C-O (-C-C-O-) which is connected to the aromatic benzene ring. The signal at 113 ppm is characteristic NMR signal for nitrile groups as well as the carbon of the phenyl ring to which the nitrile groups are connected. The presence of four different types of alkyl carbons with/without hydrogen appeared as broad NMR signals in between 23-57 ppm is mainly due to the high temperature synthetic procedures. Authors highlighted the -CH_3 carbon NMR signal by light green color at 23.54 ppm to display the presence of hydrogen in the framework structures. The obtained NMR signals of those alkyl carbons also support the retaining of frameworks on the reported CTFs structures. Additionally, it is also observed from **Supplementary Figure 7a** that with the increase of synthetic temperature from 400-600 °C, the characteristic NMR signals of specified carbons became weak, supporting the increase extent of carbonization/graphitization at elevated temperature (**Figure 3g-i**).

Supplementary Figure 7b. Solid-state ^{13}C CP-MAS NMR spectrum of CTFs (TP4).

Supplementary Figure 7a. Solid-state ^{13}C CP-MAS NMR spectra of TP4, TP5, and TP6 CTFs.

Authors additionally perform the XPS analysis of C1s configurations of the reported CTFs as shown in **Supplementary Figure 8**. The deconvolution of C1s XPS spectra by six peaks also supports the presence of six different types of chemically non-equivalent carbons as stated earlier. The deconvoluted XPS peak at 283.56 eV is related to the carbon attached with hydrogen. The others peaks at 283.86, 284.16, 284.51, 286.09, and 289.58 eV are corresponding to the presence of five different carbon configurations in the CTFs polymer as C-C, C=C, C=N, C=C-C, and C=C-O functional moieties respectively. The existence of those carbons also support the integrity of CTF structures at higher temperature up to 600 °C.

Supplementary Figure 8. Deconvoluted XPS spectra of C1s configuration of (a) TP4, (b) TP5, and (c) TP6 CTFs.

In addition, the detection of elemental proton (hydrogen) on CHN analysis (**Supplementary Table 1**), appearance of O1s peak on XPS survey analysis (**Supplementary Figure 9f**) and observed uniform distribution of oxygen on TEM elemental mapping of TP6 (**Supplementary Figure 9d**) give straightforward indication about the retention of frameworks on the reported CTFs in spite of partial carbonization/graphitization.

Supplementary Table 1. Elemental (CHNO) analysis of synthesized PIM-1 based CTFs (TP4, TP5, and TP6).

CHN-O Analyses		% C	% O	% H	% N
Theoretically		74.06	13.15	4.14	8.64
Experimentally	TP4	80.40	7.60	2.70	9.30
	TP5	80.80	6.20	1.60	11.40
	TP6	81.90	5.90	1.80	10.40

Supplementary Figure 9. Elemental composition analyses of CTFs. (d) HRTEM elemental mapping of oxygen for TP6; and (f) XPS survey spectra of TP4, TP5, and TP6 CTFs.

Authors acknowledge to the reviewer for this important comment that improves our manuscript desperately. Authors now thoroughly revised the manuscript and supporting information by adding the valuable data as stated and highlighted by red color.

[Reviewer #1's Comment (2)] *In the capacitance calculation, the authors calculate the capacitance by integrating current across the voltage range. This can be problematic because the current crosses zero. A simple integration will result in a wrong area. I strongly suggest the authors calculate the area by integrating the current difference over the voltage range and divide it by two (because there are two separate processes of charging and discharging). The reviewer is sorry for being persistent on this matter. Because there are too many wrongly reported numbers in the literature, I hope the authors agree to double-check this matter to keep the community in check.*

[Answer to Reviewer #1's Comment (2)] Authors are greatly thankful to the reviewer for his valuable suggestion and comment. Authors followed the specified method for the calculation of area by integrating the current difference over the voltage range and divide it by two. Authors found that the obtained data for specific capacitance are similar to the previously reported values. Here, authors also like to mention that the repeatability of electrochemical performances at three different scan

rates (0.01V/s, 0.05V/s, and 0.1V/s) is additionally performed to cross-check the reported capacitance data. The obtained results for three consecutive measurements in terms of specific capacitances are summarized in the **Supplementary Table 2** in the supporting information section as shown below and **Figure 4e** in the main manuscript is modified by considering the average values of specific capacitances. It is observed that high scan rate reduces the capacitance value which is quite common in microporous organic polymer electrode materials.²

Supplementary Table 2. Repeatability of electrochemical CV analysis for TP4, TP5, and TP6 CTFs in aqueous and non-aqueous electrolytes as different scan rates up to three consecutive cycles.

		Specific capacitance (F/g)											
		Scan rate (V/s)											
Electrolyte	CTFs	0.01				0.05				0.1			
		Test-1	Test-2	Test-3	Ave.	Test-1	Test-2	Test-3	Ave.	Test-1	Test-2	Test-3	Ave.
H ₂ SO ₄ , 1(M)	TP6	525	529	513	522	476	465	468	470	452	457	449	453
	TP5	306	296	303	301	257	268	261	262	231	234	245	237
	TP4	221	228	219	223	210	207	203	206	187	194	185	189
KOH, 1(M)	TP6	340	331	339	337	317	313	321	317	292	307	294	298
	TP5	245	238	251	244	225	213	215	218	207	201	197	202
	TP4	148	137	145	143	120	134	126	127	109	122	117	116
EMIM-BF ₄ (0.5M)	TP6	463	471	468	467	441	438	449	443	411	417	419	416
	TP5	317	307	313	312	295	284	289	289	258	274	263	265
	TP4	162	166	175	167	135	147	138	140	133	124	132	129

Regarding new measurements of specific capacitance listed in Supplementary Table 2, newly added two authors (Saewoong Oh and Sanghee Nam) did experiments independently to double check reproducibility and repeatability of the data.

Authors' Responses to Reviewer #4's Comments

[Reviewer's Overall Comment] *The authors have addressed my concerns partially. However, the author didn't fully answer one of my questions: "For TP4, TP5, and TP6 CTFs, what reason to cause different properties?" As the authors mentioned, "partial carbonization during the synthesis of materials at elevated temperature", then how to avoid the influence of different carbon content caused by high temperature on performance? In the manuscript, the authors have attributed to the different structures of CTFs caused by different temperatures totally. The authors should give more experimental data to eliminate the contents of carbon.*

[Response to Reviewer 4's Overall Comment] Authors are greatly thankful to the reviewer for the important comments on partial carbonization and its influence on the electrochemical and actuation performances. I recommend that the reviewer carefully check **[Answer to Reviewer #1's Comment (2)]**. And the following comments will be helpful in understanding the importance of TP6.

The reported XPS data (**Figure 3g-i**, shown below) corresponding to $N_{\text{graphitic}}$ peak of TP4, TP5, and TP6 CTFs reveal the truth of partial carbonization/graphitization. At the same time, it is also remarked that extent of graphitization increases with the increase in synthetic temperature from 400 to 600 °C. In this context, authors like to assure that this phenomenon is quite normal in CTFs research field, because graphitization/carbonization process initiated due to the increase of irreversible C-C bond formation by the reaction of cyano moieties with the increase of synthetic temperature. And the resultant CTF structures with partial graphitization having definite surface area, pore size and heteroatoms doping are controllable and reproducible due to confine synthetic procedure.³ It is also reported that the CTF structure containing graphitic nitrogen is beneficial towards the increase of electrical conductivity, which in turn enhances the electrochemical performances, such as electric double-layer capacitance.

Figure 3. Physicochemical and structural characterizations of as-synthesized novel CTFs electrode materials. (g-i) XPS N1s spectra of (g) TP4, (h) TP5, (i) TP6 materials. Display three N configurations after deconvoluted as olive: N_{triazine} , blue:

N_{cyano} , and violet: $N_{\text{graphitic}}$. Increased in $N_{\text{graphitic}}$ peak area from TP4 to TP6 reveals frameworks containing graphitic nitrogen increases with the increase in synthetic temperature.

In this study, the specific surface area and extent of graphitization apparently increases from TP4 to TP6 CTFs as the synthetic temperature rises from 400 to 600 °C. As a result, both of the electrochemical and actuation performances were enhanced from TP4 to TP6 CTFs due to the increase of electrolyte-accessible surface area and graphitization induced electrical conductivity. The increase of electric double-layer capacitance in both of the aqueous and non-aqueous electrolytes from TP4 to TP6 CTFs due to the combine effect of surface area, electrical conductivity and heteroatoms content is well reflected in the **Figure 4a-c** as shown below.

Figure 4. Electrochemical performances of synthesized electrode materials. Cyclic voltammetry (CV) analysis of as synthesized novel CTFs TP4, TP5, and TP6 in (a) aqueous 1.0 M H₂SO₄, (b) 1.0 M KOH electrolytes, and (c) non-aqueous 0.5 M EMIM-BF₄/acetonitrile ionic-liquid electrolyte solutions at a scan rate of 0.01 V/s.

Correspondingly, TP6PP ionic soft actuator shows superior actuation performances over TP5PP actuator, which again shows higher performances than TP4PP soft actuator. The electrical conductivity data of PP, TP4PP, TP5PP, and TP6PP electrodes as reported in the main manuscript (**Inset of Figure 4h**, shown below) supports the contribution of graphitization structures towards the increase of electrical conductivity. Because TP6 CTFs having relatively high graphitization content when interacted with PEDOT-PSS during the fabrication of TP6PP electrode shows highest electrical conductivity (60.88 S cm⁻¹) amongst TP5PP (34.71 S cm⁻¹) and TP4PP (13.32 S cm⁻¹) electrodes when compared to the pristine PP electrode (5.88 S cm⁻¹). Authors added this key parameter in the revised manuscript (page 14, 17).

Authors are also pointed out here that the content of carbon in the forms of condensed aromatic benzene/triazine rings in the reported CTFs is not present as impurity, they are the part of CTF structures and significantly improving the electrical conductivities of the reported polymers.

Figure 4. (h) Inset: Changes of electrical conductivity/resistivity of PP electrode upon interaction with TP4, TP5, and TP6 CTFs.

At this stage, we would like to mention that the reviewer's comments were so highly pertinent and helpful that we could enhance the quality of the article and deepen the overall level of our scientific understanding. We enormously appreciate the reviewers for their careful reviews, and we hope that with such significant improvement the paper is now acceptable for publication. All of the changes in the manuscript are marked in BLUE.

References

1. Dey, S., Bhunia, A., Esquivel, D. and Janiak, C. Covalent triazine-based frameworks (CTFs) from triptycene and fluorene motifs for CO₂ adsorption. *J. Mater. Chem. A*, **4**, 6259-6263 (2016).
2. Kou, Y., Xu, Y., Guo, Z. and Jiang, D. Supercapacitive Energy Storage and Electric Power Supply Using an Aza-Fused π -Conjugated Microporous Framework. *Angew. Chem. In. Ed.* **123**, 8912-8916 (2011).
3. Li, Y., Zheng, S., Liu, X., Li, P., Sun, L., Yang, R., Wang, S., Wu, Z. S., Bao, X. and Deng, W. Q. Conductive Microporous Covalent Triazine-Based Framework for High-Performance Electrochemical Capacitive Energy Storage. *Angew. Chem. In. Ed.* **130**, 8124-8128 (2018).

REVIEWER COMMENTS

Reviewer #1 (Remarks to the Author):

The authors have addressed my concerns and I recommend it for publication.

Reviewer #4 (Remarks to the Author):

The authors have revised the manuscript to an acceptable level. I would like to recommend the manuscript for publication, given that the authors further clarify the following points.

1) Authors attributed the better performance to the increase of electrolyte-accessible surface area and graphitization induced electrical conductivity, then what's the role of polymer? So, can the authors provide the performance data of TP7 (700oC or even higher temperature that contain more carbon) to see if no polymer is better or not.

2) What we concerned about is the structure activity relationship of the materials that prepared in this manuscript. So if the materials is still polymer or doped carbon materials is very important. The authors had provide more evidence to help explain that there are still polymers remain in the product material, such as proton NMR, sNMR, XPS, but we still have some doubts.

(1) t still very strange that in figure S2a, the residual of TP4, TP5, TP6 is 70%, 85%, 90%, if most of the materials are polymer, why there are so big difference in the weight loss of TP4 with the other two? Maybe some data like (element analysis; TGA) of TP7 is help to confirm the structure at different temperature.

(2) Usually, the NMR signal of triazine ring carbon is near 170ppm (J. Am. Chem. Soc. 2020, 142, 15, 6856–6860), even in the reference S1 of this manuscript (it is 164ppm and 172ppm), but in this materials it is 161 ppm and the signal is very weak, if it is really triazine ring carbon?

(3) No peak of cyano group in FTIR (figure 3d), but Ncyano in XPS of figure 3g-3i and the content seems very high? Why?

Authors' Responses to Reviewer #1

[Reviewer #1 (Remarks to the Author)]

The authors have addressed my concerns and I recommend it for publication.

[Response to Reviewer #1's remarks]

Authors are greatly thankful to the reviewer. We believe that the reviewer's comments were greatly helpful in enhancing the quality of this research work.

Authors' Responses to Reviewer #4

[Reviewer #4 (Overall remarks to the Author)]

The authors have revised the manuscript to an acceptable level. I would like to recommend the manuscript for publication, given that the authors further clarify the following points.

[Response to Reviewer #1's overall remarks]

Authors are grateful to the Reviewer for careful review of our manuscript and positive responses on it. Following the suggestion, we did our best by performing essential experiments to improve the quality further. All comments have been addressed carefully and incorporated in the manuscript and supplementary information as mentioned below.

[Comment (1)]: *Authors attributed the better performance to the increase of electrolyte-accessible surface area and graphitization induced electrical conductivity, then what's the role of polymer? So, can the authors provide the performance data of TP7 (700 °C or even higher temperature that contain more carbon) to see if no polymer is better or not.*

[Answer to Comment (1)]:

Authors thank to the reviewer for this comment as it really helps to clarify the importance of CTF structures for better electrochemical and actuation performances compared to high temperature carbonized structures. Following the reviewer suggestion, authors performed the electrochemical CV analysis of TP7 materials in the same non-aqueous EMIM-BF₄/CH₃CN (0.5M) solution at the scan rate of 0.01 V/s and found the **average specific capacitance value of 138 F/g**, which is lower than that of TP4 to TP6 CTFs (**Supplementary Table 2**). To check the actuation performances of TP7 as a TP7PP soft ionic actuator, authors followed the same procedure for the preparation of soft ionic actuators (**Materials and Methods**). The comparative actuation performances in terms of bending displacement of TP7PP

AWIS actuator with TP4PP, TP5PP, and TP6PP actuators under a sine wave input with peak voltage of 0.5 V and excitation frequency of 0.1 Hz are shown in **Figure R1**.

Figure R1. Actuation performances of CTF-based AWIS actuators. Time-dependent bending responses of PP, TP4PP, TP5PP, TP6PP, and TP7PP soft actuators under a sine wave input with peak voltage of 0.5 V and excitation frequency of 0.1 Hz.

It is clearly observed from the figure that although TP7PP AWIS actuator showed better performances than the PP actuator by showing larger bending displacement under same input electric stimuli and excitation frequency, but lower than that of TP4PP, TP5PP, and TP6PP actuators. These results can evidently be used to explain the importance of framework structures having partial graphitization for both of higher electrochemical and actuation performances in comparison to only carbonized materials (TP7). The relatively lower performances of TP7 is mainly due to the decrease of electrolyte ions accessible surface area of TP7 (already reported in the main manuscript, **page 9-10**) by the loss of framework structures at 700 °C [R1].

[Comment (2)]: What we concerned about is the structure activity relationship of the materials that prepared in this manuscript. So if the materials is still polymer or doped carbon materials is very important. The authors had provide more evidence to help explain that there are still polymers remain in the product material, such as proton NMR, sNMR, XPS, but we still have some doubts:

(i) *It still very strange that in figure S2a, the residual of TP4, TP5, TP6 is 70%, 85%, 90%, if most of the materials are polymer, why there are so big difference in the weight loss of TP4 with the other two? Maybe some data like (element analysis; TGA) of TP7 is help to confirm the structure at different temperature.*

[Answer to Comment 2(i)]:

Authors are thankful to the reviewer for arising this point. The different percentages of weight loss as mentioned in **Figure S2a** for TP4, TP5, and TP6 CTF polymers is mainly due to the presence of dissimilar graphitic contents. The initial weight loss up to 110 °C for TP4 is due to loss of moisture and adsorbed water molecules mainly increased the difference in total weight loss with the other two CTFs (TP5 and TP6). It has been clearly mentioned in the main manuscript (**Page 13-14**) that the reported CTF (TP4, TP5, and TP6) structures contain condensed aromatic benzene/triazine-rings as graphitic frameworks to some extent. And, the presence of those graphitic structures increases from TP4 to TP6 CTFs as there is a relevant increase of synthetic temperature from 400 to 600 °C. As a result, in TGA analysis (**Supplementary Figure 2a**) we got different percentages of residual weights for TP4, TP5, and TP6 CTFs after heating those polymers up to 800 °C, shown below as **Figure R2a**. Because the graphitic structures are more stable at higher temperature in comparison with PIM-1 frameworks having relatively lower stable ether-linkage fuse-rings (**Supplementary Figure 10b**). Amongst TP4, TP5, and TP6 CTFs, TP4 was synthesized at lowest temperature of 400 °C and accordingly it contains least amount of graphitic configurations and high percentage of ether-linkage fuse-rings. Therefore, it shows highest order of weight loss (~30%) when heated up to 800 °C temperature under N₂ atmosphere during thermogravimetric analysis, including 5-7% initial weight loss due to evaporation of moisture and adsorbed water at 110 °C. However, increase of graphitic content in TP5 and TP6 CTFs (due to higher synthetic temperature) reduces the weight loss up to ~15 and ~10%, respectively where almost none or negligible initial weight loss corresponding to moisture and adsorbed water are observed.

As suggested by the reviewer, authors additionally performed elemental analysis (by using CHNO Elemental Analyzer) and thermogravimetric analysis (by using High Resolution TGA Analyzer) of TP7 CTFs (700 °C synthetic temperature) to further confirm the degradation of framework structures or carbonization at elevated temperature. The obtained results from elemental analysis for TP7 are shown below in **Table R1** and highlighted by red colour. It is clearly observed from the Table that the carbon content increases almost 10% while the oxygen and nitrogen contents reduce sharply up to 5 and 4% respectively for TP7 materials in comparison to the TP4 CTFs. These results confirm the degradation of main backbone framework structures of CTF polymers at elevated temperature of 700 °C as the decomposition of covalent triazine frameworks is started from 650 °C (in our case it is 670 °C according to DTG plot, **Supplementary Figure 2a**), shown below as a reference [R1].

TGA spectra of TP7 material as synthesized at 700 °C is shown in **Figure R2b**. As observed from the figure there is no further major weight loss or decomposition at elevated temperature (up to 800 °C), which further confirm the decomposition of triazine frameworks above 650 °C. And, as the synthetic temperature of TP7 is more than its frameworks decomposition temperature, the TP7 material is already decomposed during synthesis.

Table R1. Elemental (CHNO) analysis of synthesized PIM-1 based CTFs (TP4, TP5, TP6, and TP7).

CHN-O Analyses		% C	% O	% H	% N
Theoretically		74.06	13.15	4.14	8.64
Experimentally	TP4	80.40	7.60	2.70	9.30
	TP5	80.80	6.20	1.60	11.40
	TP6	81.90	5.90	1.80	10.40
	TP7	90.55	2.57	1.68	5.20

Theoretical calculations were done by considering one repeating unit of CTFs.

Formula: $C_{180}H_{120}N_{18}O_{24}$

Formula weight: 2918.88 (100%)

Elemental Analysis (Wt.%): C, 74.06; H, 4.14; N, 8.64; O, 13.15

Figure R2. Thermogravimetric analysis. (a) TG-DTG spectra of PIM-1, TP4, TP5, and TP6. (b) TGA spectra of TP7.

[Comment (2)]: (ii) Usually, the NMR signal of triazine ring carbon is near 170ppm (J. Am. Chem. Soc. 2020, 142, 15, 6856–6860), even in the reference S1 of this manuscript (it is 164ppm and 172ppm), but in this materials it is 161 ppm and the signal is very weak, if it is really triazine ring carbon?

[Answer to Comment 2(ii)]:

Authors thank to the reviewer for this comment and the reference as mentioned. Authors found the reference useful for supporting the decomposition of covalent triazine frameworks above 650 °C and incorporated in the revised manuscript.

Authors are grateful to the reviewer for providing us an opportunity to elucidate the obtained ^{13}C NMR signal of triazine ring carbon of the reported novel CTFs. We are delighted to explain about the obtained solid-state ^{13}C CP-MAS NMR signal of triazine ring carbon at 161.81 ppm (**Supplementary Figure 7b**) for the CTFs. It is evident from the literature that ^{13}C NMR signal for ideal **1,3,5-triazine molecule** is appeared at 166 ppm [R2] which is shifted either in downfield or upfield according to the change of surrounding chemical and physical environments. In details, change of chemical structure, synthetic procedure, and arrangement of frameworks along with their physicochemical interactions in solid state alter the electronic configuration of triazine ring carbon of CTFs. For example, recently a two-dimensional triazine structures showed a broad signal at 142-162 ppm corresponding to carbon atoms of triazine rings in ^{13}C solid-state CP-MAS NMR [R3]; **CTF-HUST** showed ^{13}C NMR signal at 165.4 ppm for triazine ring carbon [R4]; **benzimidazole-linked CTFs** showed the same triazine ring carbon signal at 152.6 ppm [R5]; **a triazine and a keto functionalized nonmetal based COF** showed doublet ^{13}C NMR signal at 163.2 and 166.8 ppm for carbons of the triazine ring [R6]; **fluorinated CTF** materials as mentioned on the ongoing comment showed the corresponding NMR signal at 169.7 ppm [R1].

The triazine ring carbon of reported PIM-1 based CTFs having distinctive chemical structures experiences unique physicochemical interactions and correspondingly it shows the NMR signal at 161.81 ppm.

Based on the above mentioned literature reports, authors proposed a mechanism that can explain about the shifting of NMR signal for triazine ring carbon in CTFs (**Table R2**). As mentioned in the last row of **Table R2**, the reported PIM-1 based CTFs show ^{13}C NMR signal at 161.81 ppm for triazine ring carbon due to the presence of PIM-1 frameworks which partially decrease the C=N polarity of triazine unit. The **weaker NMR signal** for the triazine ring carbon is mainly obtained due to ionothermal synthesis. Because it requires higher synthetic temperature which causes partial fragmentation or graphitization [R5].

Table R2. Proposed mechanism for the shifting of solid-state CP-MAS ^{13}C NMR signal of CTFs.

Triazine unit	δ (ppm), ^{13}C	Possible reason
	166	Ideal C=N of triazine molecule.
 R=H, alkyl, X	>166	Increased the C=N polarity of triazine molecule by +R-effect of aromatic benzene ring.
 R= any -R-effect group	<166	Decreased the C=N polarity of triazine molecule by reducing the +R-effect of aromatic benzene ring.
 PIM-1-based CTFs	161.81	Presence of PIM-1 frameworks partially decreases the +R effect of adjacent aromatic benzene ring towards triazine unit. Therefore, the triazine carbon of it shows lower chemical shift (161.81 ppm) than the ideal triazine molecule.

[Comment (2)]: (iii) No peak of cyano group in FTIR (figure 3d), but Ncyano in XPS of figure 3g-3i and the content seems very high? Why?

[Answer to Comment 2(iii)]:

Authors thank to the reviewer for his comment. It is evident from literature that the appearance of cyano groups in FT-IR spectra for a CTFs indicates a lower degree of trimerization reaction whereas almost or totally disappearance of this peak corresponds to the higher degree of trimerization reaction [R1, R5, R7]. In this manuscript the observed FT-IR transmittance peak intensity of cyano ($-\text{C}\equiv\text{N}$) functional groups at 2241 cm^{-1} for PIM-1 is relatively lower than the other obtained characteristic peaks (**Figure 3d**). After successful trimerization reaction during ionothermal process, intensities of those FT-IR peaks reduce in such a way that the presence of cyano peak is not visible in the FT-IR spectra of CTFs. But, the presence of available cyano functional groups in CTFs in terms of N1s configuration can be observed in XPS analysis. Because, XPS analysis not only provided the broad survey about the elemental composition of the materials (**Supplementary Figure 9f for PIM-1 based CTFs**), but also the bonding characteristics with other elements including chemical states. It is a very powerful and sensitive tool by which one can scan a specific region of interest to get all the possible configurations of an element present in the materials. Therefore, although the presence of cyano functional groups

of the reported CTFs in this manuscript is not observable by the FT-IR analysis, they can easily be detected by the XPS through regional scanning of nitrogen configuration as N_{cyano} (**Figure 3g-i**). This observed fact is quite common for CTFs materials as reported elsewhere [R1, R8], as shown below.

R8. (a) FT-IR and (b) N1s XPS spectra of TCNQ-CTFs: **Reference:** *Angew. Chem. In. Ed.* **130**, 8124-8128 (2018).

R1. (a) FT-IR and (b) N1s XPS spectra of CTF-1-AB Stacking: **Reference:** *J. Am. Chem. Soc.*, **142**, 6856-6860 (2020).

References

R1. Yang, Z., Chen, H., Wang, S., Guo, W., Wang, T., Suo, X., Jiang, D.E., Zhu, X., Popovs, I. and Dai, S. Transformation Strategy for Highly Crystalline Covalent Triazine Frameworks: From Staggered AB to Eclipsed AA Stacking. *J. Am. Chem. Soc.* **142**, 6856-6860 (2020).

R2. Braun, S. and Frey, G. Hochauflösungs- ^{13}C -NMR-Spektroskopie. I-Analyse der

Spektren des 1, 2, 4- und 1, 3, 5-Triazins sowie ihrer Methyl- und Phenyl-derivate. *Org. Magn. Reson.* **7**, 194-198 (1975).

R3. Faghani, A., Gholami, M.F., Trunk, M., Müller, J., Pachfule, P., Vogl, S., Donskyi, I., Li, M., Nickl, P., Shao, J. and Huang, M.R. Metal-Assisted and Solvent-Mediated Synthesis of Two-Dimensional Triazine Structures on Gram Scale. *J. Am. Chem. Soc.* **142**, 12976-12986 (2020).

R4. Wang, K., Yang, L.M., Wang, X., Guo, L., Cheng, G., Zhang, C., Jin, S., Tan, B. and Cooper, A. Covalent triazine frameworks via a low-temperature polycondensation approach. *Angew. Chem. Int. Ed.* **56**, 14149-14153 (2017).

R5. Tao, L., Niu, F., Wang, C., Liu, J., Wang, T. and Wang, Q. Benzimidazole functionalized covalent triazine frameworks for CO₂ capture. *J. Mater. Chem. A*, **4**, 11812-11820 (2016).

R6. Bhadra, M., Kandambeth, S., Sahoo, M.K., Addicoat, M., Balaraman, E. and Banerjee, R. Triazine functionalized porous covalent organic framework for photo-organocatalytic E-Z isomerization of olefins. *J. Am. Chem. Soc.* **141**, 6152-6156 (2019).

R7. Mukherjee, S., Das, M., Manna, A., Krishna, R. and Das, S. Dual strategic approach to prepare defluorinated triazole-embedded covalent triazine frameworks with high gas uptake performance. *Chem. Mater.* **31**, 3929-3940 (2019).

R8. Li, Y., Zheng, S., Liu, X., Li, P., Sun, L., Yang, R., Wang, S., Wu, Z. S., Bao, X. and Deng, W. Q. Conductive Microporous Covalent Triazine-Based Framework for High-Performance Electrochemical Capacitive Energy Storage. *Angew. Chem. Int. Ed.* **130**, 8124-8128 (2018).

REVIEWERS' COMMENTS

Reviewer #1 (Remarks to the Author):

The authors have revised the manuscript in response to my comments at an acceptable level for publication.

Authors' Responses to Reviewer #1

Reviewer #1 (Remarks to the Author)

The authors have revised the manuscript in response to my comments at an acceptable level for publication.

Response to Reviewer #1's remarks

Authors are greatly thankful to the reviewer.